# Antarctic Winds: Pacemaker of Global Warming, Global Cooling, and the Collapse of Civilizations

**W. Jackson Davis** [1,2,*] **and W. Barton Davis** [2]

1   Division of Physical and Biological Sciences, University of California, Santa Cruz, CA 95064, USA
2   Environmental Studies Institute, Santa Cruz, CA 95062, USA; WBDavis@yahoo.com
*   Correspondence: JacksonDavis@EnvironmentalStudiesInstitute.org

**Abstract:** We report a natural wind cycle, the Antarctic Centennial Wind Oscillation (ACWO), whose properties explain milestones of climate and human civilization, including contemporary global warming. We explored the wind/temperature relationship in Antarctica over the past 226 millennia using dust flux in ice cores from the European Project for Ice Coring in Antarctica (EPICA) Dome C (EDC) drill site as a wind proxy and stable isotopes of hydrogen and oxygen in ice cores from EDC and ten additional Antarctic drill sites as temperature proxies. The ACWO wind cycle is coupled 1:1 with the temperature cycle of the Antarctic Centennial Oscillation (ACO), the paleoclimate precursor of the contemporary Antarctic Oscillation (AAO), at all eleven drill sites over all time periods evaluated. Such tight coupling suggests that ACWO wind cycles force ACO/AAO temperature cycles. The ACWO is modulated in phase with the millennial-scale Antarctic Isotope Maximum (AIM) temperature cycle. Each AIM cycle encompasses several ACWOs that increase in frequency and amplitude to a Wind Terminus, the last and largest ACWO of every AIM cycle. This historic wind pattern, and the heat and gas exchange it forces with the Southern Ocean (SO), explains climate milestones including the Medieval Warm Period and the Little Ice Age. Contemporary global warming is explained by venting of heat and carbon dioxide from the SO forced by the maximal winds of the current positive phase of the ACO/AAO cycle. The largest 20 human civilizations of the past four millennia collapsed during or near the Little Ice Age or its earlier recurrent homologs. The Eddy Cycle of sunspot activity oscillates in phase with the AIM temperature cycle and therefore may force the internal climate cycles documented here. Climate forecasts based on the historic ACWO wind pattern project imminent global cooling and in ~4 centuries a recurrent homolog of the Little Ice Age. Our study provides a theoretically-unified explanation of contemporary global warming and other climate milestones based on natural climate cycles driven by the Sun, confirms a dominant role for climate in shaping human history, invites reconsideration of climate policy, and offers a method to project future climate.

**Keywords:** Antarctic Centennial Oscillation (ACO); Antarctic Centennial Wind Oscillation (ACWO); Antarctic Isotope Maxima (AIMs); Antarctic Oscillation (AAO); Bond Cycle; climate forecasting; climate hindcasting; Dansgaard–Oeschger Oscillations; empirical climate projection; glacial expansion; glacial retreat; Heinrich events; Little Ice Age; Medieval Warm Period

## 1. Introduction

Global temperature measured instrumentally has increased by ~0.8 °C since 1880 [1,2], showing that the Earth has warmed since the end of the Little Ice Age (LIA) in 1860. The cause of this global warming signal, however, is under debate. According to the Anthropogenic Global Warming (AGW) hypothesis,

the current global warming signal is caused largely or entirely by the accelerated emission of greenhouse gases such as carbon dioxide ($CO_2$) into the Earth's atmosphere from human activities [3–6]. The AGW hypothesis holds that global temperature will continue to increase through an enhanced "greenhouse" effect if the concentration of $CO_2$ in the Earth's atmosphere continues to rise [3–6].

Recent empirical evidence shows, however, that on multimillennial timescales over most of known climate history atmospheric $CO_2$ concentration was not correlated with, and therefore did not cause, global temperature change [7]. Atmospheric $CO_2$ concentration is correlated with temperature on the millennial timescale of recent glacial cycling, but multiple studies [8–11] suggest that during deglaciation, temperature change leads $CO_2$ change, i.e., $CO_2$ does not initiate glacial termination. Feedbacks to temperature from $CO_2$ are believed to amplify warming during deglaciation, but modeling studies with MODTRAN demonstrate that $CO_2$ feedbacks are strongest at the lowest atmospheric concentrations characteristic of initial glacial termination [7]. As $CO_2$ concentration increased during deglaciation and thereafter to post-industrial highs, marginal radiative forcing from $CO_2$ declined exponentially [7]. Atmospheric $CO_2$ therefore exerted small and diminishing causal influences on temperature during late glacial termination and even smaller effects at the higher $CO_2$ concentrations that prevail today.

An alternative hypothesis to explain contemporary global warming holds that the observed increase in global temperature is caused by a natural climate cycle that is currently reaching a periodic peak. According to this Natural Global Warming (NGW) hypothesis, Earth's climate is intrinsically cyclic across all timescales [12,13] and the planet is now experiencing a transient warm phase that will soon reverse. The most likely candidate for the causal climate cycle is the recently discovered [12] Antarctic Centennial Oscillation (ACO). Diverse evidence [13] suggests that the ACO is the evolutionary precursor of, and is identical with, the contemporary and well-known Antarctic Oscillation (AAO). We therefore abbreviate this natural temperature cycle using the combined acronyms, ACO/AAO.

The ACO/AAO has oscillated continuously for at least the last 226 thousand years (Ky), as far back in time as extant sampling resolution can detect centennial-scale cycles in the Vostok paleoclimate temperature-proxy record [12]. Individual ACO/AAO temperature-proxy cycles summate ("piggyback") and facilitate temporally (grow larger over time with repetition) across millennial-scale time periods to culminate in recurrent Antarctic Isotope Maxima (AIMs), the proximate source of Dansgaard–Oeschger (D-O) oscillations of 5–8 °C recorded in Greenland ice cores [12,13]. Because the ACO/AAO induces such large, regular and rapid temperature increases in the Northern Hemisphere (NH), this natural temperature cycle exhibits orders-of-magnitude greater temperature forcing power (up to ~0.8 °C/year) than is required in principle to contribute in whole or part to the much smaller and slower global warming signal of ~0.8 °C over the 140 years since 1880 [1,2] (0.006 °C/year).

With this background, our study has three purposes. The first is to test the hypothesis that ACO/AAO cycles are driven by relaxation oscillation (RO) of Westerly Winds (WWs) in the Southern Ocean (SO) [12,13]. Our findings confirm a key prediction of the RO hypothesis by showing that every ACO/AAO temperature-proxy cycle is associated with a wind-proxy cycle of the ACWO at all of eleven major drill sites in Antarctica over every time period investigated. We find that the ACWO is in turn modulated in phase with the millennial-scale Antarctic Isotope Maximum (AIM) temperature cycle. The interaction of these two natural climate cycles explains numerous well-known global climate milestones, including the current global warming signal, the Medieval Warm Period (MWP), the LIA, and their previous recurrent homologs. This data-driven climate model further explains the mechanisms underlying these and other landmark climate events, including the AIM temperature-proxy cycle, consequent D-O oscillations recorded in Greenland ice cores, correlated Heinrich events in the NH, and the coupled Bond Cycle and corresponding Bond events in the NH. Finally, circumstantial evidence suggests that these natural climate cycles are forced by millennial-scale variation in solar output associated with the Eddy cycle of sunspot activity.

The second purpose of this study is to develop and apply a new, quantitative, empirical tool for projecting global climate based on the ACWO. We find that the ACWO wind-proxy cycle documented

here hindcasts accurately the recurrent MWP/LIA cycle, severe droughts associated with the LIA and its homologs, and the corresponding collapse of the 20 largest human civilizations of the last four millennia. The hindcasting skill of the ACWO model supports its use in climate forecasts, which project the same climate pattern that has repeated for the past 226 millennia of Earth's climate history: imminent global cooling associated with the coming negative phase of the ACO/AAO, continuing with three centuries of global temperature oscillation accompanied by increasing net warming that represents the recurrent homolog of the next MWP, and concluded with the recurrence of the next LIA homolog in ~4 centuries.

The third purpose of this study is to weigh evidence on the cause of the contemporary global warming signal in the context of the formal theory of causation based on the classic counterfactual framework. When adjudicated within this conceptual framework, the preponderance of empirical evidence suggests to us that contemporary global warming is caused less by atmospheric $CO_2$ and more by a natural climate cycle, the ACO/AAO, which is in turn driven by the ACWO wind cycle introduced here.

The unprecedented climate scenario projected by the ACWO wind cycle has implications for climate science and policy. In respect to climate science, global climate projections based on the ACWO wind cycle provide a simple, empirical falsifying test to distinguish between the anthropogenic and natural global warming hypotheses. Whereas theoretical computer models based on the AGW hypothesis project that continuous global warming will accompany future increases in greenhouse gas concentration in the atmosphere [3–6], the data-driven ACWO wind cycle model based on the NGW hypothesis and formalized here projects the opposite, imminent global cooling, irrespective of the concentration of $CO_2$ in the atmosphere. The ACWO wind cycle offers a way to test these hypotheses conclusively while simultaneously providing an empirical tool for accurately projecting future global climate change on decadal and centennial scales.

In respect to climate policy, global warming that arises even partly from natural rather than anthropogenic causes requires revising the contemporary climate policy regime, which is based on the premise of human culpability for a warming planet. Even if all of the contemporary global warming signal is natural in origin, however, limiting anthropogenic emissions of $CO_2$ remains a policy priority given the possible role of atmospheric $CO_2$ and consequent ocean acidification in past mass extinctions of biodiversity [7].

## 2. Methods

This study uses methods, data, and analytical approaches identical to those reported in our previous papers [7,12,13] and particularly their Supplementary Material (SM) [12,13]. We refer to those studies for the details of methods summarized in this section.

### 2.1. Data and Rationale

Temperature-proxy records were sourced and ACO temperature-proxy peaks labeled as reported [12–15]. Temperature at Vostok was computed from deuterium excess using the conversion constant of 9‰/°C [14] (p. 431). Temperature proxies from the remaining ten Antarctic drill sites studied here are sourced as cited [13]. The temperature records used here are specific to the indicated drill sites, and therefore, each is shaped significantly by local meteorology and especially proximity to the ocean [13]. We therefore regard temperature signals at Antarctic drill sites as indirect and sometimes delayed indicators of temperature signals at the source of the ACO/AAO 1500 km off the east coast of Antarctica [13]. We use individual temperature records because to our knowledge an integrated temperature record for all of Antarctica does not exist and might not be helpful given the well-known temperature differential between East and West Antarctica.

Dust flux is widely used as a proxy for wind velocity and has been evaluated at several drill sites in Antarctica [16–21]. In contrast to the use of multiple, site-specific temperature proxies, we used dust flux from a single drill site, the European Project for Ice Coring in Antarctica (EPICA) Dome C (EDC), as a wind proxy. The rationale for this selection is the strong evidence that the wind-proxy

records at drill sites distributed widely across Antarctica are "remarkably similar" to each other and to those recorded in the Andes [22], [23] (p. 256), as would be expected for a high-velocity, vortical, mixed laminar/turbulent airflow like the WWs. In this case, it is conceptually immaterial which drill site is used for sampling wind-proxy data. We therefore used the longest and best-resolved dust-flux record, EDC, as a proxy for winds at all drill sites in Antarctica. Given the rapid movement of wind over long distances and the circular structure of the Antarctic Circumpolar Vortex, we infer that the wind (dust-flux) signal at each drill site, including EDC used here, is similar or identical to the wind signal at every other drill site and at the source of the ACO/AAO in the SO [13].

Dust-flux records analyzed here from EDC were sampled originally at high resolution, up to several dust sample datapoints per year for the most recent millennium of the Holocene declining to 1 datapoint per 10 years by 226 thousand years before 1950 (Kyb1950) [16]. Such high-resolution sampling of dust-flux record extends back in time more than 800 Kyb1950 [16]. This long and highly-resolved dust-flux record establishes the EDC paleoclimate dust-flux record as the most precise and lengthy indicator available as a proxy of Antarctic wind, supplemented by an equally long and well-resolved temperature record [24]. We averaged high-resolution dust-flux samples [16] in 40-60-year bins to estimate changes in Antarctic wind velocity. The date assigned here to each mean wind-proxy value is the midpoint of the 40-60-year bin-width over which the dust-flux proxies were averaged. Data on the timing of the beginnings and ends of past civilization are compiled at [25] based on [26–30], to which we added the dates of collapse of the early [31] and late [32] Persian Empires.

## 2.2. Analytic Protocols and Programs

As in our previous studies [7,12,13], we compared different paleoclimate records, in this case temperature proxy and dust-flux records, using a standardized approach consisting of quasi-quantitative cross-record matching of cycle peaks followed by quantitative auto- and cross-correlation analysis (time domain) and spectral analysis (frequency domain). Labeling of Vostok temperature-proxy cycles started with the most recent cycle, which peaked 149 years before 1950 (Yb1950) and is labeled ACO/AAO cycle 1 (12). Peak matching of temperature cycles at Vostok with those at all other drill sites [12] was achieved by first identifying the nearest available homologous peaks in other temperature records and labeling them with the same cycle numbers as Vostok [12,13]. The use of "signpost" climate events whose time of occurrence is accepted consensually within the climate science community enabled unambiguous cross-matching of temperature peaks in paleoclimate records from different drill sites [12,13].

Initial matching of temperature-proxy peaks with wind-proxy peaks at EDC was done following the same procedure. Each previously-identified and numbered temperature-proxy cycle was matched with the closest nearby wind-proxy (dust flux) cycle. Over several time frames in the records analyzed, the cross-record coherence of their peaks was 1:1, i.e., matching was deterministic. In some cases, smaller peaks in the wind-proxy record were not matched with corresponding temperature peaks (<5% of all wind-proxy peaks). These few anomalous wind cycles are highlighted in all figures by enclosing them in red ovals and interpreted in terms of underlying mechanisms and forces.

Following peak matching as above, we did quantitative, progressive (lagged) auto-correlation on the same temperature and wind records and progressive lagged cross-correlation between them. Auto-correlation is done by shifting a time-series record relative to itself one datapoint at a time in successive "lag orders" and computing the Pearson product moment correlation coefficient for each shift. Lag order is therefore defined as the number of datapoints by which a dataset was shifted relative to itself and is therefore dimensionless, i.e., it lacks units. Lag order is converted to time by dividing the duration of the paleoclimate record analyzed by the number of lag orders in the auto-correlogram to yield time equivalence in terms of years per lag order. Statistically discernible lagged auto-correlation coefficients in any auto-correlogram signify significant periodicity in the corresponding time series record at the frequency measured by the number of lag orders and the time equivalence between peaks or troughs in auto-correlograms.

Progressive or lagged cross-correlation is operationally identical to auto-correlation, except rather than correlating a time series with itself to test for periodicity, one dataset (wind) is correlated with a different dataset (temperature) to explore both periodicity and the degree of cross-coupling between the two variables. Matching of times in the wind record was done in most cases by averaging data so that time increments correspond to the temperature record, as required for matched-pair cross-correlation. In rare cases, nearby temperature proxies were matched with wind peaks, in which case the differences in chronology were $<\pm1\%$. As in the case of auto-correlation, lag order in cross-correlograms describes the number of datapoints by which the datasets compared are shifted relative to each other. Time equivalence in cross-correlograms is again computed as the record duration in years divided by the number of lag orders. Statistically discernible cross-correlation demonstrates that both variables are significantly periodic and that they oscillate at the same frequency, estimated from cross-correlograms from the number of cycles in the cross-correlogram over the time period analyzed. The mean repetition period of the corresponding cycles is approximated as the duration of the record analyzed divided by the number of cycles contained within it. Conversely, if two periodic time series demonstrate discernible cross-correlation, then, by definition, their respective cycles are coupled, i.e., matched 1:1, in which case the cross-record coherency as used here and defined previously [12,13] approximates 100%.

Confidence limits for all auto- and cross-correlation coefficients were computed by calculating the smallest Pearson correlation coefficient that is discernible at $p < 0.05$ for the sample size associated with every fifth or tenth lag order (one-sided $t$-tests), depending on the record duration (greater spacing for longer records). Connection of these points using a sixth-order polynomial equation as a best-fit smoothing function following the method of least squares yielded the dashed blue lines shown in all correlograms in this paper as 95% confidence limits for corresponding correlation coefficients.

We did spectral analyses on the corresponding wind-proxy and temperature-proxy records and compared them quantitatively using the same approach published previously [7,12,13]. Spectral analysis of dust-flux and temperature records was done with SAS JMP software, version 12.2.0. This commercially-available code returns the probability that frequency spectra were generated by white (Gaussian or random) noise using Fisher's Kappa (k) statistic [12]. This statistic and the corresponding probabilities of nonrandom distribution of the spectral density profile are reported in each pertinent figure caption. Most spectral analyses were done on climate records that were first linearly detrended, following our earlier approach [12], with the exception of the dust-flux record in Figure 13, where the spectral density profile was computed from non-detrended time series data because the quasi-parabolic shape of the corresponding time series rendered linear detrending meaningless.

Comparison of spectral frequencies between independent paleoclimate records was quantified as described previously [7,12,13]. Every peak in one spectral density periodogram was matched with the nearest visible peak in the other, and each such peak was identified on spectral periodograms by arrows. The matches are highlighted by enclosing the shafts of matching arrows in red ovals in all such figures (Figures 9 and 13). Multiple spectral peaks were identified for both wind-proxy and temperature-proxy records. Peaks in one record were paired with the nearest peak in the other record and the percent difference between them was computed. Quantitative differences between the respective matched frequency peaks were computed in both relative and absolute terms. Relative means were computed using signs of individual datapoints and are less conservative because positive and negative values canceled to form smaller mean error values. Absolute means are computed without regard to sign and are considered a more accurate measure of variance.

We integrated both temperature and dust-flux proxies over successive century-long periods in an effort to develop a possible aggregate indicator of Antarctic temperature. Integration was achieved using two methods. The first consisted of computing the area beneath the time-series curves of temperature and dust-flux in 100-year increments using superimposed best-fit triangles to estimate the area under the temperature curve. The second method consisted of integrating (summing) area by weighing the cut-out area beneath the respective curves for 100-year periods using a decigram

Fusion digital weighing scale. The repeat-measure error of best-fit triangle method trials was <±1%, while the repeat-measure weighing error was <±5%. The two methods gave comparable conclusions. We present only the latter.

*2.3. Limitations*

Previously we identified multiple periodicities in cross-correlograms that correspond to comparable rhythms in the original climate data [7,12,13]. Here we quantify these periodicities in order to estimate the cycle period manifest in coupled paleoclimate records for comparison with measured values over the same time periods. These quantifications are approximate, however, since they are based in some cases on relatively small modulation of correlation coefficients whose differences would not themselves reach statistical discernibility. The centennial-scale periodicities visible in cross-correlograms, indicated by open double-headed arrows in the corresponding graphs, nonetheless match periods measured over the same time periods from paleoclimate records on average within ±2–4% (relative and absolute measuring error, respectively).

Our study is restricted to the calmest wind periods of the last 226 millennia of paleoclimate history, i.e., the flat regions between the windier epochs associated with the Great Ice Ages (GIAs) (MISs). We selected these less windy periods for analysis because they encompass the most recent climate, the Holocene, and because ACWO cycles are easier to detect visually and analyze quantitatively in such relatively calm periods. Numerous similar calm periods punctuate climate history for the past 226 millennia, comprising about half the corresponding paleoclimate record. Analysis of the intervening windier periods is more difficult and generally beyond the scope of this study, although we were able to quantify one relatively windy period at 149 Kyb1950 as described below. It is possible and perhaps likely that the wind pattern identified and analyzed in the present study is overridden or at least modified by the stronger forces and consequently greater winds associated with GIAs.

**3. Results and Discussion**

We begin with an overview of the relationship between wind and temperature proxies at a single drill site, EDC. We then extend the analysis using the wind-proxy record from EDC and the temperature-proxy records from ten additional major Antarctic drill sites, with comparable results. The bulk of our analysis follows, using EDC wind data and temperature data from nearby Vostok. Within each section the results are organized by time period, beginning with the most recent paleoclimate records.

*3.1. Wind and Temperature at EDC*

Comparison of the temperature-proxy record at EDC with the wind-proxy record at the same drill site over the last four millennia shows that both wind and temperature are periodic on a millennial scale. The pattern is clearest visually in the wind-proxy record, where wind velocity oscillates on a centennial scale that is modulated on a millennial scale. Centennial-scale wind cycles grow larger over millennial timescales and then drop precipitously at 3.3, 1.8, and 0.7 Kyb2013 (Figure 1a, brown curve). Temperature at EDC shows a similar but more variable pattern (Figure 1a, red curve).

During the last 4000 years, every centennial-scale peak in the temperature record at EDC is accompanied by a corresponding peak in the wind record (Figure 1a). Therefore, cross-record peak coherence from temperature to wind is 100%, i.e., temperature peaks match 1:1 with wind peaks. One small wind cycle is unmatched by any corresponding peak in the temperature record, yielding a cross-record coherency of 96.6%. The mean wind-to-temperature peak coherency averaged for both directions, therefore, is 98.3%, comparable to the ~95% mean coherency between temperature peaks in records from different drill sites, following the definitions and methodology of cycle coherence detailed in [13].

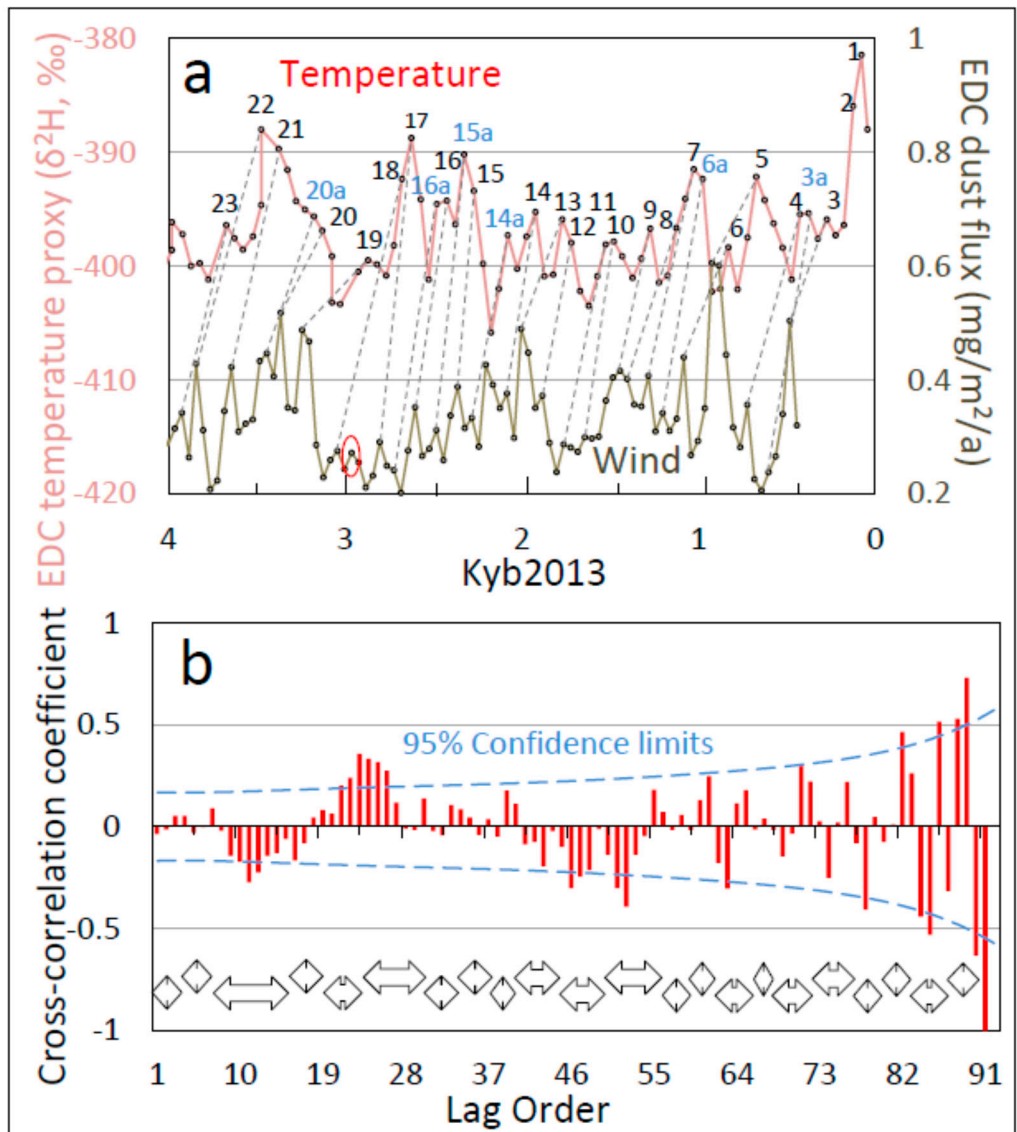

**Figure 1.** One-to-one coupling between identified temperature-proxy oscillations at EPICA Dome C (EDC) and dust flux (a wind proxy) at EDC for the last four millennia. (**a**) 1:1 correspondence between wind-proxy cycles and identified temperature-proxy cycles, connected by dashed lines to assist visualization of cross-record coherence. Each wind peak corresponds 1:1 with the indicated temperature peak. EDC temperature peaks are labeled following the numbering protocol developed for corresponding peaks at Vostok [12,13] (**b**) Cross-correlation between temperature proxies and dust flux at EDC over the period 4000-375 years before 2013 (Yb2013). The dashed blue lines in part b represent 95% confidence limits (positive and negative) for the value of the correlation coefficient associated with different lag orders (sample sizes). Double-headed arrows demarcate the underlying centennial-scale periodicity determined from the cross-correlation record. The cross-correlation is statistically discernible in most (52.4%) cycles, ranging in magnitude from weak to moderate.

Matching peaks between wind and temperature as in Figure 1a is quantitative in that it synchronizes climate events in the one record with the closest available temporal match in the other, and based on the timing of nearby well-known ("signpost") climate events (see Section 2 and [12,13]). Quantitative corroboration is provided by progressive ("lagged") cross-correlation of the wind and temperature records, in which the two records are aligned in time and then shifted past each other one datapoint at a time and the correlation coefficient computed at each corresponding shift (lag order).

The relationship between wind and temperature is assessed quantitatively from two metrics: periodicity in the cross-correlogram at the same frequency as the correlated cycles, and statistically discernible cross-correlation coefficients in the computed correlograms (see Section 2).

Applying these criteria to the EDC wind/temperature data reveals that the cross-correlation is periodic at the approximate frequency of the ACO/AAO cycles (double-headed arrows in Figure 1b). Of the 21 cycles in the cross-correlation record, 11 (52.4%) contain either a peak or trough that exceeds statistical discernibility at $p < 0.05$ (dashed blue curves in this and subsequent figures). Visible centennial-scale periodicity occurs at a calculated period of 162.0 years, similar to the measured ACO/AAO period over this same time period of 173.9 years (−6.8% difference). We conclude that peaks in temperature and wind are matched 1:1 at EDC, and that the cross-correlation between the records is statistically discernible, ranging from weak to moderate.

To determine whether 1:1 matching of wind and temperature peaks at EDC occurred during earlier and colder periods of paleoclimate history, we examined the time period from 71-63 Kyb1950 (Figure 2), corresponding to near the Last Glacial Maximum (LGM) prior to the Last Glacial Termination (LGT) and the transition to the warmer Holocene. This time period encompasses the well-studied [12] AIM 19 and AIM 18 (Figure 1). The cross-record peak coherence between the wind and temperature records from EDC during this time period is 100% in both directions, i.e., every wind-proxy peak is matched 1:1 with a temperature-proxy peak, and conversely. Applying the same quantitative criteria as done above for the Holocene (Figure 1) shows that 8/12 (66.7%) of visible cycles in the cross-correlogram (double-headed arrows in Figure 2) contain a peak or trough that exceeds statistical discernibility at $p < 0.05$, corresponding to weak-to-moderate cross-correlation coefficients. The 12 visible cycles are distributed over 6857 years corresponding to 53 lag orders. Each lag order therefore corresponds to 129.4 years (6857/53). The estimated period of these cycles is therefore 551 years while the period of ACO/AAO temperature peaks at Vostok is measured as 212.0 years [13].

On the basis of the difference between estimated and measured period, we conclude that the cross-correlation between wind and temperature over this time period is statistically discernible but weak despite the 1:1 correspondence between wind and temperature peaks. This weak correlation stems in part from the dominance in this temperature-proxy record of the longer periodicity of 1454 years, corresponding to the AIM cycle (filled arrows in Figure 2b), and is not unexpected given that temperature at individual drill sites is an indirect measure of temperature at the source of the ACO/AAO as described in the Sections 2 and 4.

The criterion for identifying AIM events in earlier studies was limited to highly visible summated ACO/AAO cycles in the temperature record. At EDC, however, additional AIM-like events are visible in both the wind and temperature records (Figure 2a, putative wind cycles labeled 18a-18d). These additional cycles are generally clearer in the wind record, as might be expected given that temperature records are indirect measures of distant events at the source of the ACO/AAO and shaped by the local meteorology of different drill sites. The additional AIM cycles visible in the wind record (Figure 2a) suggest that AIM cycles may be more frequent and ubiquitous than previously believed (see Section 4). Support for this hypothesis is provided below based on spectral analysis.

## 3.2. Wind at EDC versus Temperature Across Eleven Antarctic Drill Sites

The above analysis of wind at EDC versus (v.) temperature at EDC was repeated for wind at EDC v. temperature at ten additional major drill sites in Antarctica, with similar results as exemplified in Figure 3 and summarized in Table 1. Previously we found that every ACO/AAO cycle identified at any of these eleven drill sites occurs also at every other drill site [12,13]. We show above that every ACO/AAO temperature cycle at one drill site, EDC, is accompanied by a wind cycle. It follows that every ACO/AAO temperature cycle at each of the eleven drill sites analyzed here is accompanied by an ACWO wind cycle. Here we confirm this expectation for all eleven Antarctic drill sites evaluated in this study.

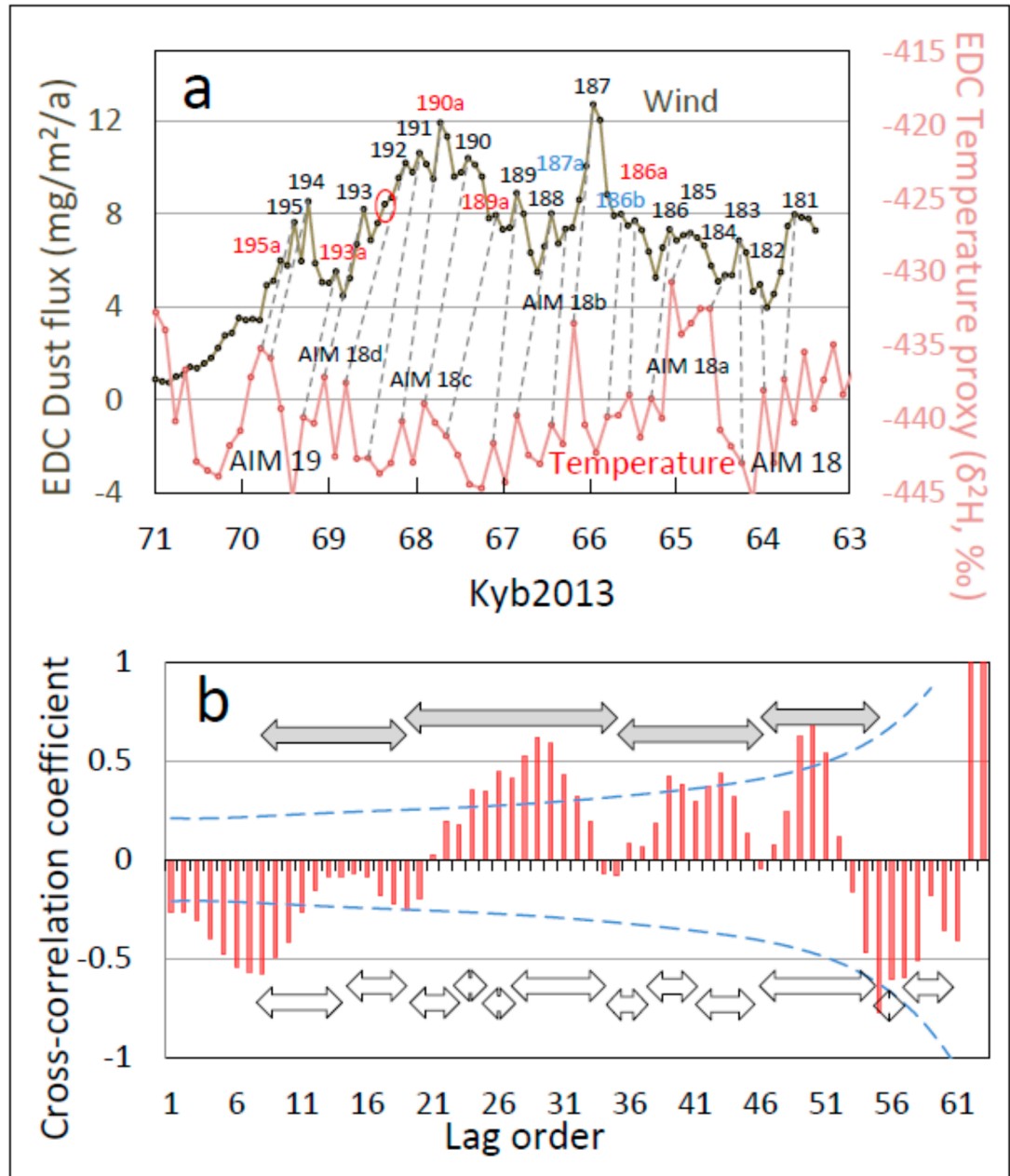

**Figure 2.** One-to-one coupling between temperature-proxy oscillations at EPICA Dome C (EDC) and dust flux (a wind proxy) at EDC for the time period from 71–63 thousand years before 2013 (Kyb2013). (**a**) 1:1 correspondence between wind-proxy cycles and identified temperature-proxy cycles. Corresponding peaks are connected by dashed lines to assist visualization of cross-record coherence. (**b**) Cross-correlation between temperature proxies and dust flux over the period 70–61 Kyb2013. The dashed blue lines in part b represent 95% confidence limits (positive and negative) for the value of the correlation coefficient associated with different lag orders (sample sizes). In this and subsequent figures, double-headed arrows demarcate the underlying periodicity visible in the cross-correlation record. Filled arrows show millennial-scale periodicity while open arrows designate cycles that recur with centennial-scale periodicity.

To illustrate this approach, cross-correlation between EDC wind and Taylor Dome temperature (Figure 3a) shows regular centennial-scale oscillations. Of the 18 centennial-scale cycles identified in Figure 3a, nine (50.0%) include at least one cross-correlation coefficient that is statistically discernible at $p < 0.05$. Therefore, both cross-correlated variables—wind and temperature—are by definition

periodic, as is visually apparent in the time series. These oscillations in the cross-correlogram occur at a frequency estimated from the open arrows in Figure 3a as 205.8 years. The measured period of the ACO/AAO at Vostok, and therefore at all other Antarctic drill sites, over the time frame from 4000-149 Yb1950 is computed from data presented in [12] as 186.0 years, 9.6% smaller. These results collectively demonstrate quantitatively that centennial-scale temperature changes at Taylor Dome are periodic and matched 1:1 with ACWO wind cycles at EDC.

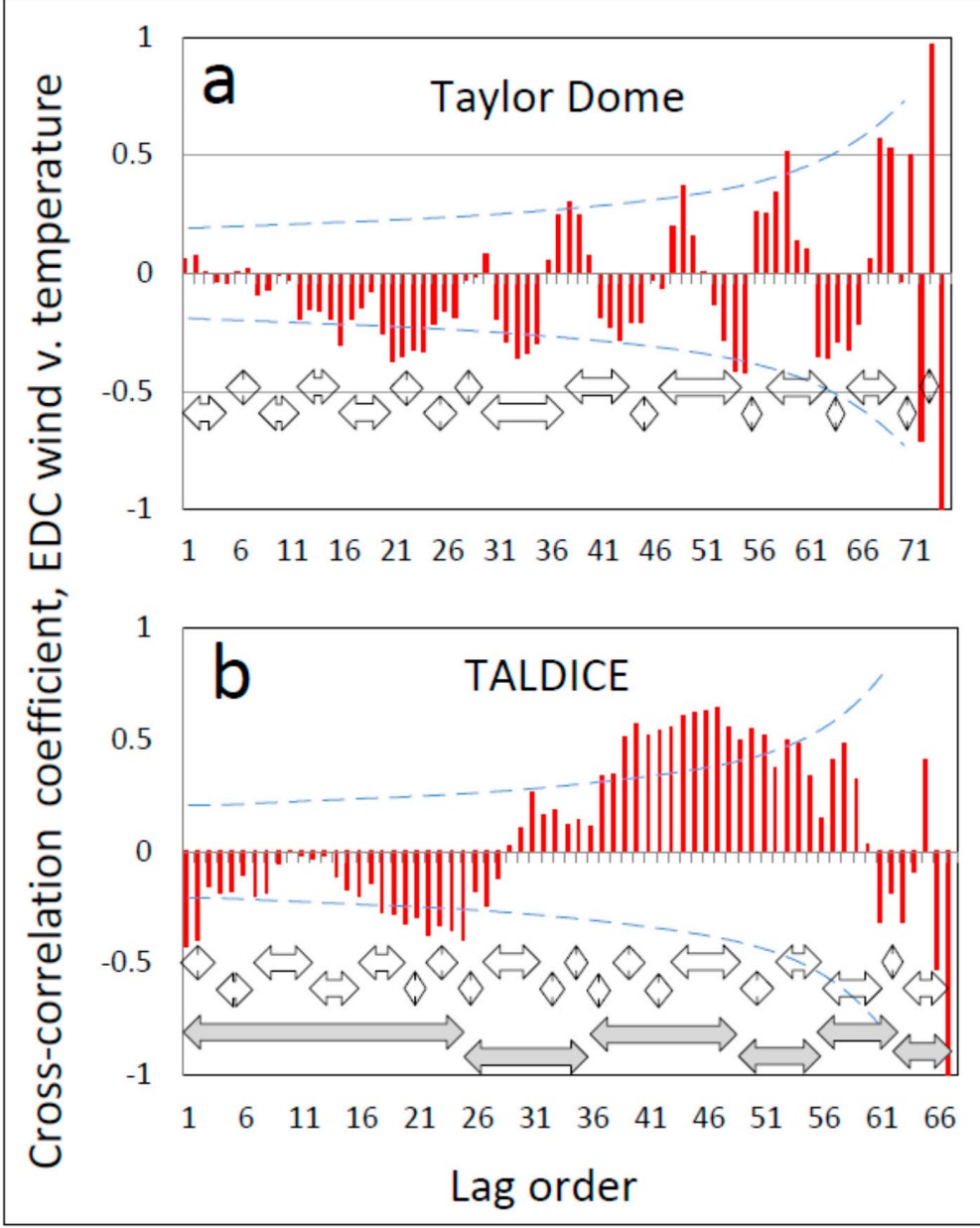

**Figure 3.** Progressive or lagged cross-correlation between wind proxies at EDC and temperature proxies at the indicated drill sites. (**a**) Cross-correlation between wind and temperature at Taylor Dome for the time period 70–61 thousand years before 2013 (Kyb2013). (**b**) Cross-correlation between wind and temperature at TALDICE for the time period from 4000–360 years before 1950 (Yb1950). The dashed blue lines in both graphs represent 95% confidence limits (positive and negative) for the value of the correlation coefficient associated with different lag orders (sample sizes). Double-headed open arrows demarcate the underlying centennial-scale periodicity visible in the cross-correlation record, while filled arrows show millennial-scale periodicity.

Similarly, at nearby TALDICE (Figure 3b), centennial-scale cycles occur in the cross-correlogram between temperature proxies and EDC wind proxies at an estimated frequency of 200.0 years (open arrows in Figure 3b), 5.7% less than the measured value of 212 years of the ACO/AAO over the same time period at Vostok [12] and therefore at all Antarctic drill sites [13]. At TALDICE, however, a second, longer periodicity in the cross-correlogram is visible at a measured period of 1266.7 years, similar to the measured period of the AIM and Bond cycles of ~1470 years [12,13] and designated by filled arrows in Figure 3b. Over all eleven Antarctic drill sites analyzed here, the longer periodicity is prominent in cross-correlograms from five drill sites (Byrd, James Ross Island, Law Dome, Siple Dome, and TALDICE,) across the Holocene, but absent or weak in the other six evaluated (EDC, Vostok, Dome Fuji, EDL, Taylor Dome, and Dome B) (Table 1).

**Table 1.** Coupling between wind proxies at EDC and local temperature proxies at eleven Antarctic drill sites as measured using lagged cross-correlation.

| 1.<br>Antarctic Drill Site (Exemplified by Figures in Parentheses) | 2.<br>Period Analyzed, Years before 1950 (Yb1950) | 3.<br>Periodicity Visible in Cross-Correlogram | 4.<br>Centennial Period Estimated from Cross-Correlogram (years) | 5.<br>Measured ACO/AAO Period over Same Time at Vostok (Years) | 6.<br>Percent Difference | 7.<br>Percent Cycles with Statistically-Discernible r Values |
|---|---|---|---|---|---|---|
| Law Dome | 13,040–9020 | Centennial and millennial | 287.1 | 212.0 | 26.2% | 64.3% (9/14) |
| TALDICE (Figure 3b) | 13,040–9020 | Centennial and millennial | 200.0 | 212.0 | −5.7% | 63.2% (12/19) |
| Siple | 13,040–9020 | Centennial and millennial | 243.8 | 212.0 | 13.0% | 62.5% (10/16) |
| Byrd | 13,040–9020 | Centennial and millennial | 211.0 | 212.0 | −0.5% | 63.2% (12/19) |
| James Ross Island | 4020–380 | Centennial and millennial | 148.0 | 186.0 | −26.2% | 44.0% (11/25) |
| EPICA Dronning Maud Land | 2043–443 | Centennial and millennial | 131.8 | 145 | −9.1% | 45.5% (5/11) |
| Dome Fuji | 3980–360 | Centennial and millennial | 162.8 | 186.0 | −12.5% | 42.9% (9/21) |
| Vostok (Figure 4b) | 2000–380 | Centennial | 200.0 | 145.5 | 27.3% | 57.1% (4/7) |
| Vostok (Figure 5) | 4020–380 | Centennial | 183.2 | 186 | −1.5% | 78.9% (15/19) |
| Vostok | 6020–380 | Centennial | 187.6 | 188 | 0.2% | 72.4% (21/29) |
| Vostok (Figure 10b) | 21,969–397 | Centennial and millennial | 432.1 | 219 | 49.3% | 69.4% (34/49) |
| Vostok (Figure 12b) | 71,315–63,395 | Centennial and millennial | 274.4 | 212 | 22.7% | 87.5% (21/24) |
| EDC (Figure 1) | 4083–443 | Centennial and millennial | 162.0 | 173.9 | −6.8% | 52.4% (11/21) |
| EDC (Figure 2b) | 70,400–63,700 | Centennial and millennial | 541.0 | 212.0 | 60.8% | 75.0% (9/12) |
| EPICA Dome B | 4005–360 | Centennial | 137.8 | 186 | −25.9% | 20.0% (5/20) |
| Taylor Dome (Figure 3a) | 4000–360 | Centennial | 205.8 | 186 | 9.6% | 50.0% (9/18) |
| **Averages** | NA | NA | NA | NA | **7.6–14.8%** | **59.3%** |

Parameters of the wind/temperature relationship across eleven Antarctic drill sites (column 1) as assessed by lagged cross-correlation between wind at EDC and temperature at the indicated drill site over the indicated time periods (column 2). Each row corresponds to a separate cross-correlation analysis. This table evaluates cross-correlograms between wind and local temperature (e.g., Figure 3) computed over the indicated time periods (column 2) in terms of the time scale of observed periodicity(ies) (column 3), the estimated period of centennial-scale cycles (column 4, calculated from the duration of open arrows in graphs (e.g., Figure 3), measured ACO/AAO period at Vostok over the same time period [12] (column 5), the percent difference between estimated and measured periods from the cross-correlograms (column 6), and the percent of cycles in the correlogram that included at least one peak or trough correlation coefficient discernible at $p < 0.05$. (column 7). The averages at the bottom of columns 6 and 7 (bold font) represent relative (computed with signs) and absolute (computed without signs) values, respectively. The average (bottom row, column 7) is skewed toward representation of Vostok because data from Vostok for several different time periods are included in the average. Abbreviations: EDC, EPICA Dome C; NA, not applicable.

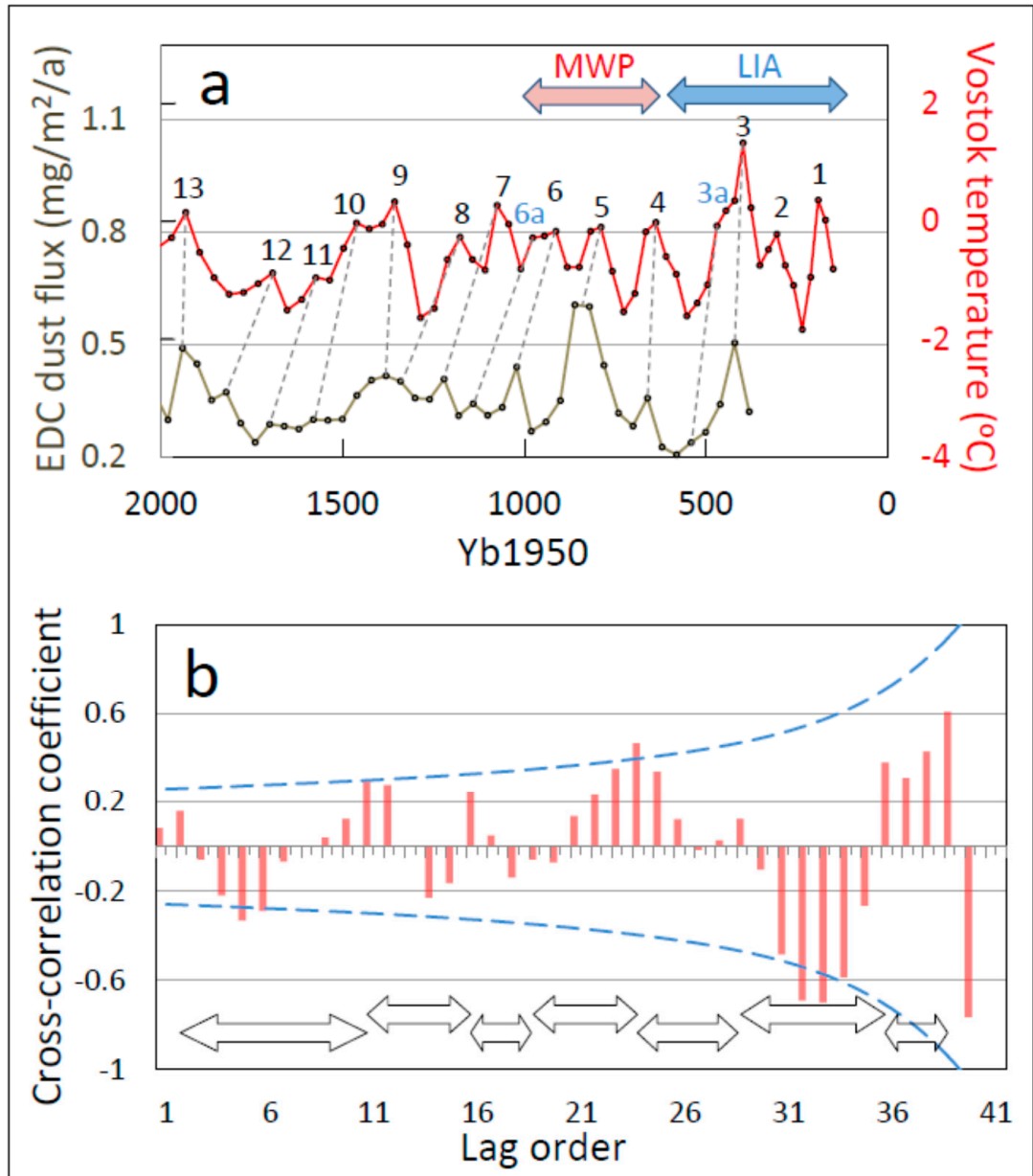

**Figure 4.** One-to-one coupling between temperature proxy oscillations at Vostok and dust flux (a wind proxy) at nearby EPICA Dome C (EDC) for the last two millennia. (**a**) 1:1 correspondence between wind-proxy cycles and identified temperature-proxy cycles. Wind and temperature peaks are connected by dashed lines to illustrate 100% cross-record coherence. (**b**) Cross-correlation between temperature proxies and dust flux over the period ~2000–397 years before 1950 (Yb1050). The dashed blue lines in part b represent 95% confidence limits (positive and negative) for the value of the correlation coefficient associated with different lag orders (sample sizes). Additional abbreviations: MWP, Medieval Warm Period; LIA, Little Ice Age.

We have no explanation for this difference in the expression of the millennial cycle at different drill sites. However, the five drill sites at which the AIM cycle is manifest more strongly in the cross-correlogram between EDC wind and temperature are all within the physical proximity of maritime influences, while the six drill sites that show little or no millennial-scale oscillation generally lie at higher, drier, and colder elevations on the EAP. Most records for older and colder time periods, e.g., the LGT, also show presence of the millennial-scale periodicity (Table 1). These results collectively

suggest a possible influence of temperature on the appearance in temperature records of the millennial period. Colder and drier locations appear to contain less representation of the millennial-scale cycle.

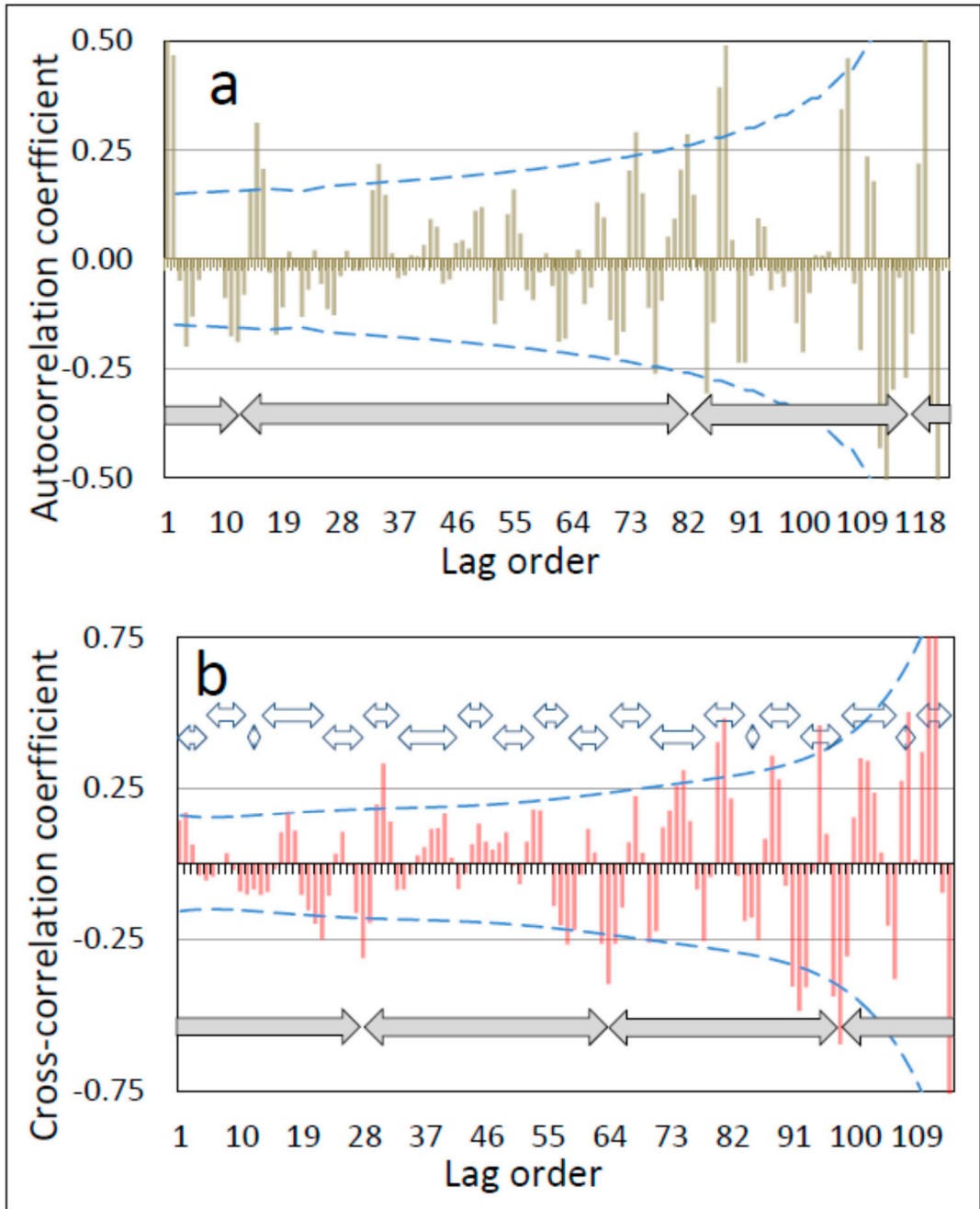

**Figure 5.** Auto- and cross-correlograms for dust-flux at EPICA Dome C (EDC, brown curve) and temperature at nearby Vostok (red curve) for the time period 4009–397 years before 1950 (Yb1950). (**a**) Discernible periodicity in lagged autocorrelation of dust flux. (**b**) Periodicity in cross-correlation between dust- and temperature-proxies over the same time period. The dashed blue lines represent 95% confidence limits (positive and negative) for the value of the correlation coefficient associated with different lag orders (sample sizes). The horizontal open double headed arrows demarcate centennial-scale repetition frequencies, while the longer, millennial-scale periodicities are visible as modulations of the correlation coefficients at periodicities of millennial scale (filled arrows; see text).

Comparable findings from all eleven Antarctic drill sites included in this study are summarized in Table 1.

Given that the ACO/AAO cycle originates ~1500 km off the east coast of Antarctica [13], it is perhaps not surprising that the wind cycle at EDC is weakly or moderately cross-correlated with the temperature cycle at every individual drill site. The temperature record at Law Dome, nearest among drill sites to the locus of ACO/AAO generation in the SO off the east coast of Antarctica [13], might be expected to provide the most accurate correlation with temperature from Antarctic drill sites, but temperature-proxy data from Law Dome across the entirety of the Holocene have not yet been published [13].

These results collectively show that centennial-scale wind peaks measured at EDC are matched 1:1 and discernibly cross-correlated with ACO/AAO temperature peaks at each of the eleven major Antarctic drill site evaluated here. We selected Vostok for further in-depth analysis in part because the temperature data have been evaluated most extensively for this drill site and in part because the millennial-scale AIM cycle is not reflected as strongly in temperature-proxy record, simplifying the analysis of the shorter centennial (ACO/AAO) cycle.

*3.3. Wind and Temperature at Vostok*

3.3.1. The Holocene

During the most recent Holocene, every temperature-proxy cycle at Vostok is accompanied by a centennial-scale wind-proxy cycle at EDC (e.g., Figure 4). The MWP coincides with temporal summation of temperature-proxy cycles at Vostok [12] to a higher mean net temperature, visible in the temperature-proxy record as increased area under the temperature curve (ACO/AAO cycles 4–8, Figure 1) and expressed globally as the four-century era of elevated temperature comprising the MWP (~900–1300 Current Era or CE). These wind-proxy cycles culminate in the largest temperature-proxy cycle of the sequence, ACO/AAO cycle 3, followed by the deepest trough, between ACO/AAO cycles 1 and 2, which coincides with the LIA that immediately follows the MWP (Figure 1). The timing of these paleoclimate temperature milestones is therefore hindcast accurately by the corresponding wind-proxy (dust-flux) record.

We assessed the strength of coupling between the wind- and temperature-proxy records quantitatively by computing the lagged cross-correlation coefficients between the respective paleoclimate records (Figure 4b). As expected under the hypothesis that these variables oscillate at the same frequency, the cross-correlation coefficient oscillates with lag order at the period of both cycles. Peak cross-correlation coefficients are statistically discernible at $p < 0.05$ in 4/7 (57.1%) cycles, demonstrating a moderate but significant periodicity in the wind- and temperature-proxy records and statistically-discernible coupling between them. This coupling in turn implies 1:1 matching between cycles in the wind and temperature records.

Expanding the time frame to 4000–397 Yb1950, lagged auto-correlation of the wind-proxy record demonstrates significant periodicity at centennial scales (Figure 5a). Lagged cross-correlation with the temperature-proxy record discloses strong coupling between the wind- and temperature-proxy records beginning at zero lag with positive cross-correlation (Figure 5a). The centennial cycle period measured from the cross-correlogram is 139.2 years, compared with the measured cycle period of wind- and temperature-proxies of 138.5 years (Table 1). This dominant frequency is modulated on a longer, millennial scale visible in the cross-correlogram, as denoted by filled double-headed arrows (Figure 5). The millennial scale of these cycles suggests that they are expressions of the AIM cycle in the SH.

We did the same analysis between wind at EDC and temperature at Vostok for the expanded time period from ~6–0 Kyb1950. Over this expanded time period, every identified ACO/AAO temperature-proxy cycle at Vostok is coupled 1:1 with an ACWO wind-proxy cycle at EDC (Figure 6). Several smaller wind cycles are not matched in the temperature record (red ovals enclosing unmatched datapoints in Figure 6), however, which is discussed further below.

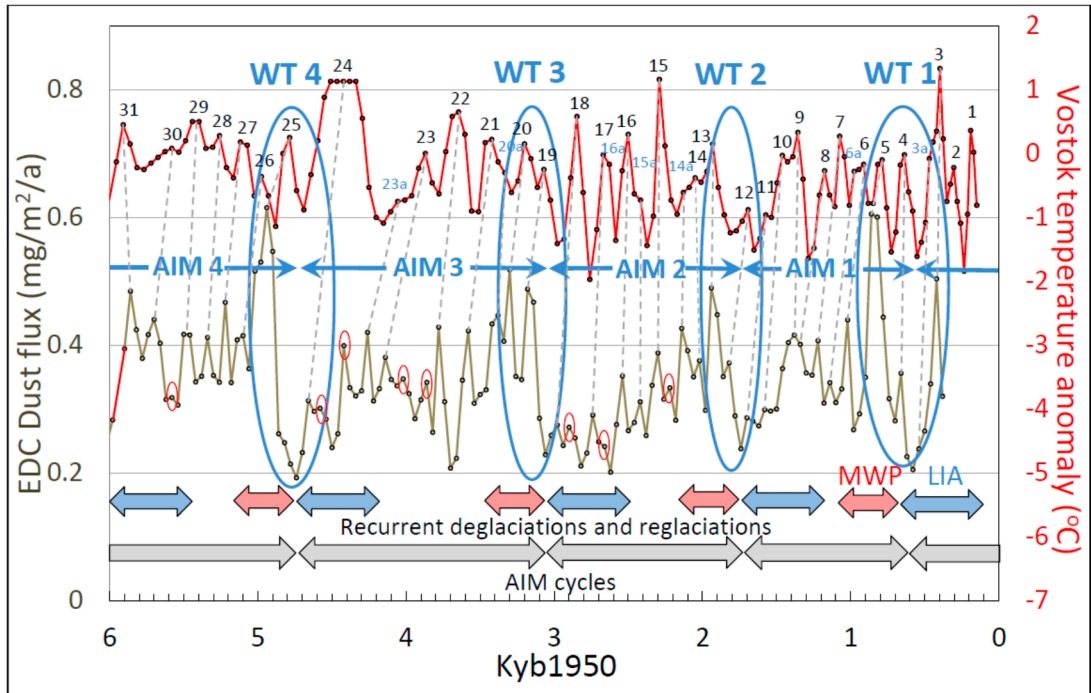

**Figure 6.** One-to-one coupling between wind proxies (dust flux) at EPICA Dome C (EDC) (lower brown curve) and temperature anomaly computed from deuterium excess at Vostok (upper red curves) over the last six millennia. Dashed lines connect wind-proxy peaks with corresponding temperature-proxy peaks, which are numbered according to the quantitative ACO/AAO cycle nomenclature developed earlier [12]. Red ovals enclose wind-proxy cycles that are unmatched in the temperature-proxy record (see text). Temperature extremes hindcasted in register with the Wind Terminus (WT) of the Antarctic Centennial Wind Oscillation (ACWO) are shown by colored bidirectional horizontal arrows (red = warm, blue = cold). The red segment of the otherwise brown wind-proxy curve at 6000–5900 years before 1950 (Yb1950) denotes a period of missing data that was filled by interpolation for visual but not analytic purposes. Additional abbreviations: AIM, Antarctic Isotope Maximum; MWP, Medieval Warm Period; LIA, Little Ice Age; WT, Wind Terminus.

The empirical wind- and temperature-proxy cycles used here to hindcast and forecast global climate are summarized in Figure 6. Centennial wind-proxy cycles that are coupled with ACO/AAO cycles are modulated with millennial periodicity in phase with the millennial-scale AIM temperature-proxy cycle. Each ACWO culminates in a Wind Terminus (WT) (enclosed by blue ovals in Figure 6), which is the largest ACWO wind-proxy cycle of the millennial sequence and includes a peak in wind velocity followed by the deepest ACWO wind-proxy trough corresponding to a maximal wind lull. The most recent WT coincided with the LIA, while previous recurrent warm and cold epochs hindcasted from the position of the WT are identified by red and blue arrows respectively at the bottom of Figure 6. These recurrent warm and cold epochs postulated from the wind pattern of the ACWO coincide closely with known periods of glacial retreat and advance over the same time periods (see below), demonstrating the skill of the ACWO wind cycle in hindcasting global temperature-proxy cycles. Effective hindcasting of past climate supports the validity of the climate forecasts presented below using this same empirical climate model.

ACO/AAO temperature-proxy cycles at different Antarctic drill sites are associated 1:1 with the largest ACWO wind cycles throughout most of the Holocene, i.e., cross-record homogeneity approaches 100% (Figure 6). In contrast, as noted above, a few smaller wind cycles in the dust-flux record do not match a corresponding temperature cycle (red ovals enclosing peak wind-proxy oscillations in Figure 6). These mismatches are not caused by averaging error, which is orders of magnitude

smaller, nor sampling resolution, which is unusually high for EDC wind-proxy data (see Section 2). The unmatched wind-proxy cycles in Figure 6 are, therefore, presumed to be non-artifactual and to reflect a genuine characteristic of paleoclimate winds in Antarctica. Such unmatched cycles are reminiscent of frequency division that characterizes relaxation oscillators [33], termed "frequency demultiplication" [34]. These unmatched wind cycles may hold clues to the external force that drives them, developed further in the Section 4.

The dates of the collapse of all human civilizations over the past four millennia have been collated recently [25] based on details available from the historical record [26–32]. Superimposing this historical record on the climate hindcasts produced here using the ACWO wind cycle reveals that every major human civilization of the past 4000 years disappeared in association with the recurrent global cooling. Of 20 major civilizational collapses tracked here, all occurred during or near the LIA or its earlier recurrent homologs (Figure 7). As developed in the Section 4, available evidence indicates that lower agricultural productivity during the LIA led to population decline and collapses of the corresponding civilizations. As also developed there, climate-driven civilizational collapse characterizes only the largest civilizations, presumably owing to the greater vulnerability of their populations to food shortages, famine, and resulting political instability.

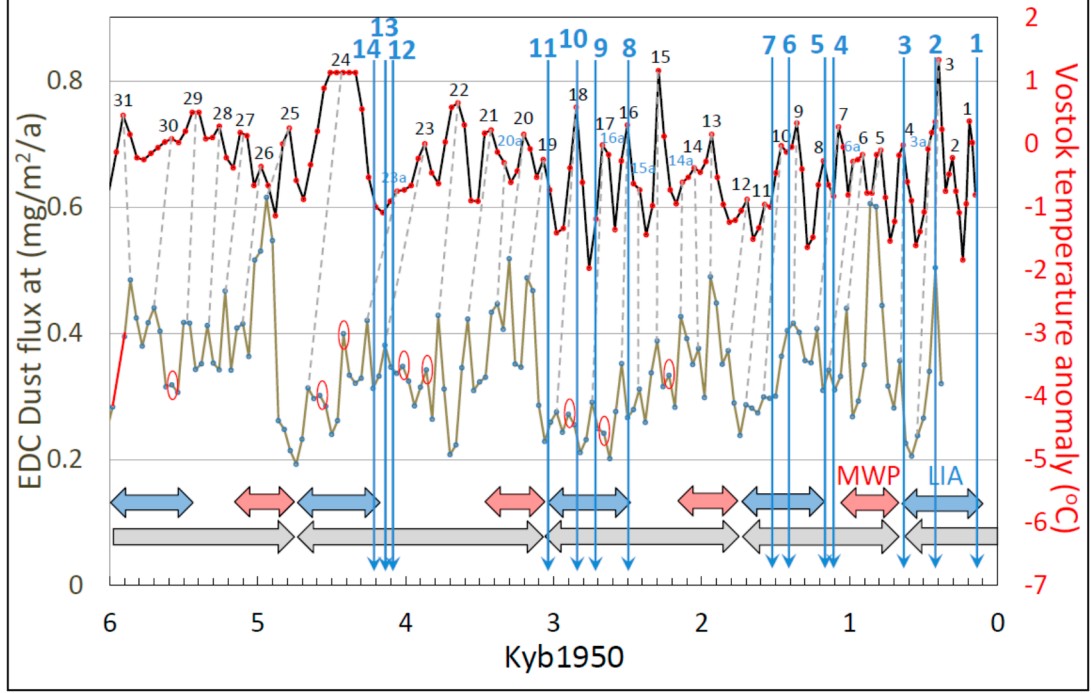

**Figure 7.** As in Figure 6, but with superimposed dates of civilizational collapses [25,31,32] indicated by numbered vertical blue arrows. The 20 largest human civilizations of the past six millennia collapsed during or near the Little Ice Age (LIA) or one of its recurrent precursors (blue double headed arrows near bottom of graph). Numbered vertical blue arrows designate the date of severe droughts and/or the collapse of identified civilizations as follows: 1, Second Persian Empire (Safavid Dynasty) [31]; 1–3, severe droughts in the U.S., fall of Cahokia (U.S.), Anasazi (U.S.), Vikings, (Greenland), Ming Dynasty (China), and Ancient Egypt III [26]; 4, Tiwanaku [26] (Bolivia); 5, Maya [26] (Central America); 6, Mochica [26] (Peru); 7, Roman Empire [27]; 8, First Persian Empire [32]; 9, Saqqaq [28] (Greenland); 10, Hittites [26] (Turkey), Ancient Egypt II [26–29]; 11, Mycenae [30] (Europe); 12, Hittites [26–29]; 13, Akkadian Empire [26] (Mesopotamia); 14, Egypt I, Mesopotamian and Indus civilizations [26,27]. Abbreviations as in Figure 6.

As noted above and illustrated in Figure 6, the MWP coincides with a period in which each temperature oscillation at Vostok does not return to the lower baseline temperature of previous

ACO/AAO cycles owing to temporal summation and facilitation of individual temperature cycles and particularly wind cycles (ACO/AAO cycles 4–8, Figure 1). Instead, each new temperature cycle begins before the previous one has ended, yielding algebraic summation of the temperature signal and the consequent increasing baseline temperature over the period of several ACO/AAO cycles. On this basis we hypothesized that the corresponding greater area under the temperature-proxy curve (the integral, or sum) is an accurate indicator of the integrated warming that is recognized as the MWP.

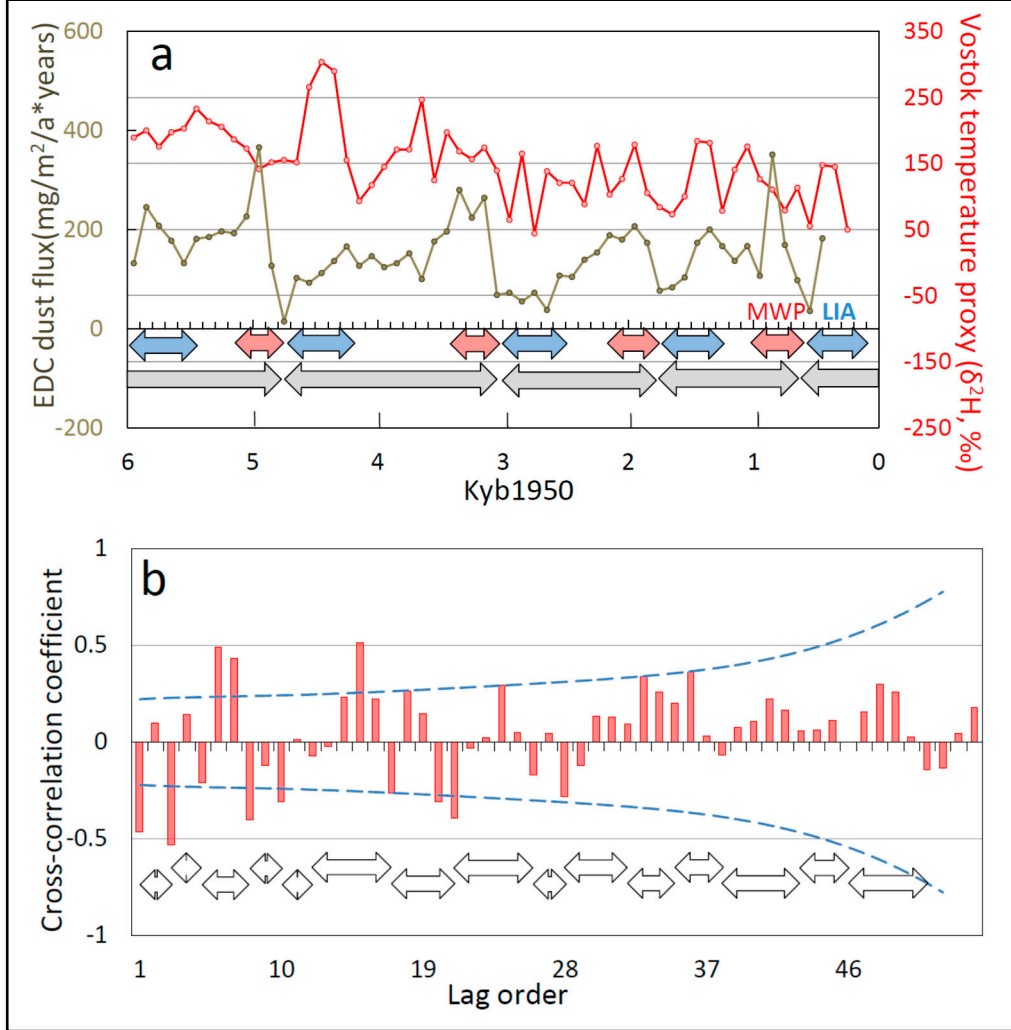

**Figure 8.** Integrated temperature-proxy and wind-proxy time series over six millennia in 100-year intervals. (**a**) relationship with recurrent warm/cold cycles of the past, including the Medieval Warm Period (MWP) and the Little Ice Age (LIA) and their earlier homologs (double-headed red and blue arrows, respectively). (**b**) Cross-correlogram of temperature-proxy and wind-proxy records showing a weak but discernible relationship, as signified by the finding that cross-correlation coefficients at cycle peaks reach significance in half the cycles. The dashed blue lines in part b represent 95% confidence limits (positive and negative) for the value of the correlation coefficient associated with different lag orders (sample sizes). Additional abbreviation: Kyb1950, thousand years before 1950.

To test this hypothesis, we integrated both temperature and wind proxies over 100-year intervals (see Section 2) and plotted them against time (Figure 8a). This exercise illustrates that integrated temperature from a single drill site (in this case Vostok) is an approximate if weak indicator of MWP timing. Recurrent warm and cold cycles are generally weakly or moderately correlated with the integrated temperature record, as shown by the relatively disorganized and statistically-weak lagged cross-correlogram between temperature and wind proxies (Figure 8b). This finding is not necessarily

unexpected, since Vostok is among the most remote of Antarctic drill sites. Temperature at drill sites closer to the source of the ACO/AAO, or perhaps integrated temperature across all drill sites, might be correlated more strongly with wind. Exploring these hypotheses further is beyond the scope of this study.

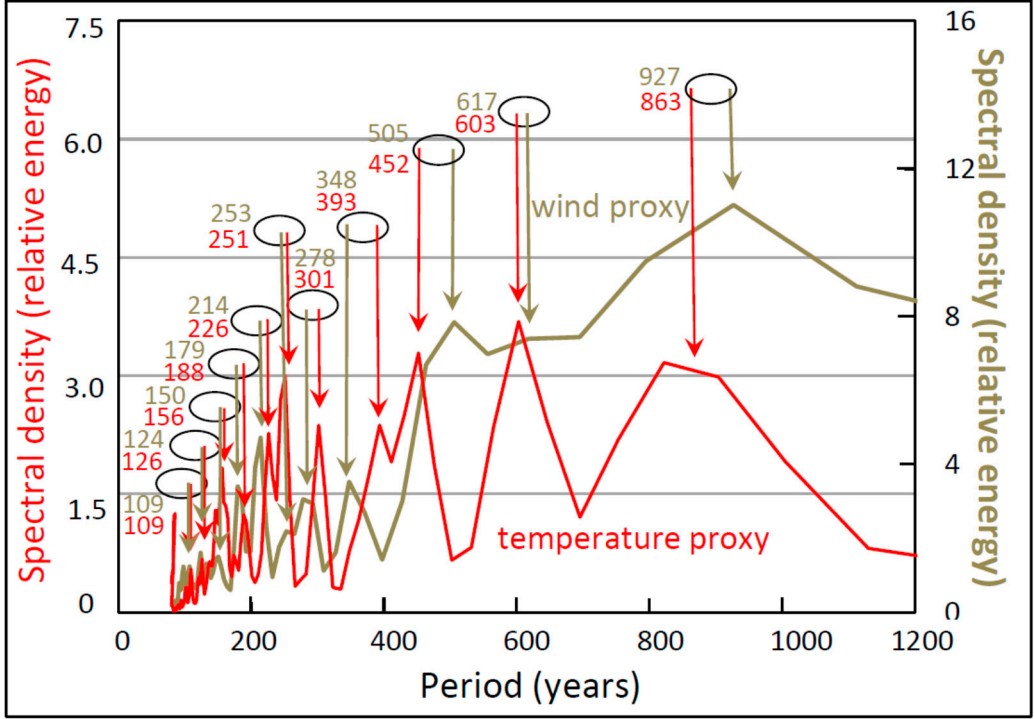

**Figure 9.** Spectral density periodograms showing similar spectral frequency profiles of dust flux at EPICA Dome C (EDC, brown curve) and a temperature proxy at nearby Vostok (red curve) for the Holocene from 6000–397 years before 1950 (Yb1950). Ovals enclose shafts of arrows that identify the most closely matched peaks at the periods in years shown beside each oval. Fischer's Kappa statistic and corresponding probability that the distribution is discernibly different from white noise are (dust-flux record) 4.95 and 0.38, and (temperature record) 17.3 and $8.7 \times 10^{-7}$. The statistical discernibility of individual spectral density peaks was not computed, although by analogy with previous research [12] (Figure 3), the largest 6–9 peaks are likely candidates for discernibility at $p < 0.05$.

In contrast to the temperature-proxy record at Vostok, integrated wind-proxies from EDC (brown curve in Figure 8a) are a strong predictor of past recurrent warm and cold periods, including the MWP and LIA. Integrated wind proxies build steadily over every AIM cycle to a peak that coincides with the corresponding recurrent warm period, followed by a steep drop that coincides with the next recurrent cold period, including the most recent LIA. Integration of wind proxies illustrates the process of temporal summation (addition or "piggybacking" of successive cycles) and temporal facilitation (growth in amplitude of cycles over time with repetition) in the wind cycle, which was documented previously in the temperature record [12] and in this study in the wind record. We conclude that integrated temperature records from individual drill sites may be less useful than integrated wind records for evaluating these and other climate phenomena.

Analysis in the time domain of the wind/temperature proxy relationship is complemented by comparisons in the frequency domain, i.e., spectral analysis of corresponding proxy records (Figure 9). As expected if the identified ACWO wind-proxy cycle drives the corresponding ACO/AAO temperature-proxy cycle, the spectral periodograms of wind- and temperature-proxies show similar frequency profiles. Identified spectral energy peaks for wind-proxies match the closest peaks in temperature proxies by ±0.6–2.9% on average (relative-absolute values) (Figure 9). Such similar

frequency profiles for wind- and temperature-proxies imply that these cycles oscillate at approximately the same frequency.

Visual inspection of the wind record during the last glaciation showed that the occurrence of AIM cycles across paleoclimate history may be more frequent than previously believed. For example, AIM 18 and AIM 19 are separated by four potential additional AIM cycles that have not been recognized previously (labeled AIM 18a–18d in Figure 2a). These putative new AIM cycles are more visible in the wind record than in the temperature record (Figure 2a). If these are genuine wind cycles corresponding to previously unidentified AIM cycles, then the spectral density profile of the wind record is expected to show a strong peak at the approximate periodicity of the AIM cycle. We confirm this hypothesis with the finding that a strong and broad spectral density peak occurs at a peak period of 927 years in the wind record, matched by a comparable spectral density peak in the temperature record of 863 years in the temperature record (Figure 9). We conclude that AIM cycles extend farther back in time than previously known.

### 3.3.2. The Last 22 Millennia

Over longer paleoclimate records there are sufficient datapoints (wind- and temperature-proxy peaks) to enable meaningful regression of the amplitude of temperature-proxy cycles against the amplitude of corresponding Antarctic wind cycles (Figure 10). This regression reveals a non-linear relationship that is best fit (method of least squares, $R^2 = 0.7531$) by the following logarithmic function:

$$y = 2.167\ln(x) - 3.4084 \qquad (1)$$

Equation (1) and the underlying empirical data imply that over low wind velocities (0–5 mg/m$^2$/a of dust flux), peak temperature drops rapidly with increasing wind velocity. At higher wind velocities (5–25 mg/m$^2$/a of dust flux), temperature-proxy peaks remain relatively constant in amplitude as wind velocity increases (Figure 10a). We cannot exclude the possibility that this non-linear relationship reflects, in whole or part, an artifact resulting from a non-linear relationship between wind velocity and dust flux. The cross-correlogram between wind-proxies and temperature-proxies shows tight coupling over this range (Figure 10b), with most peak correlation coefficients discernible at $p < 0.05$.

The wind/temperature proxy relationship depicted in Figure 10 presents an apparent dilemma, namely, negative and positive correlations between wind and temperature each manifest when computed over different time periods. The negative correlation quantified in Figure 10 is evident qualitatively in time series (Figure 11), where large increases in dust flux are coupled with corresponding declines in temperature. In contrast, and as documented previously [12,13], the smaller increases in wind-proxy amplitude that pace individual ACO/AAO cycles are positively correlated with temperature-proxies. This dilemma may have a straightforward resolution. Stronger winds are presumably driven by global jet streams governed by the equator-to-pole heat gradient, while the smaller wind cycles that generate individual ACO/AAO cycles are the consequence of local, internal relaxation oscillation in which wind presumptively drives regional (Antarctic) and global temperature.

By the time the LGT has transitioned to the warmer Holocene, wind peaks decline to the smallest amplitude seen for the 226-millennium duration of the paleoclimate record analyzed here (Figure 11). This period of relative calm may have a straightforward interpretation: the transition from the LGM to the LGT is accompanied by a strong increase in solar insolation at 65° N, which warms the poles and therefore reduces the equator-to-pole heat gradient. Consequently, Antarctic wind velocities are expected to decline to historically low levels, as observed (Figure 11). This empirical finding implies that although contemporary climate is sometimes portrayed as extreme, the wind-proxy record shows the opposite: the present climate is the calmest period in the paleoclimate record for at least the past few hundred millennia.

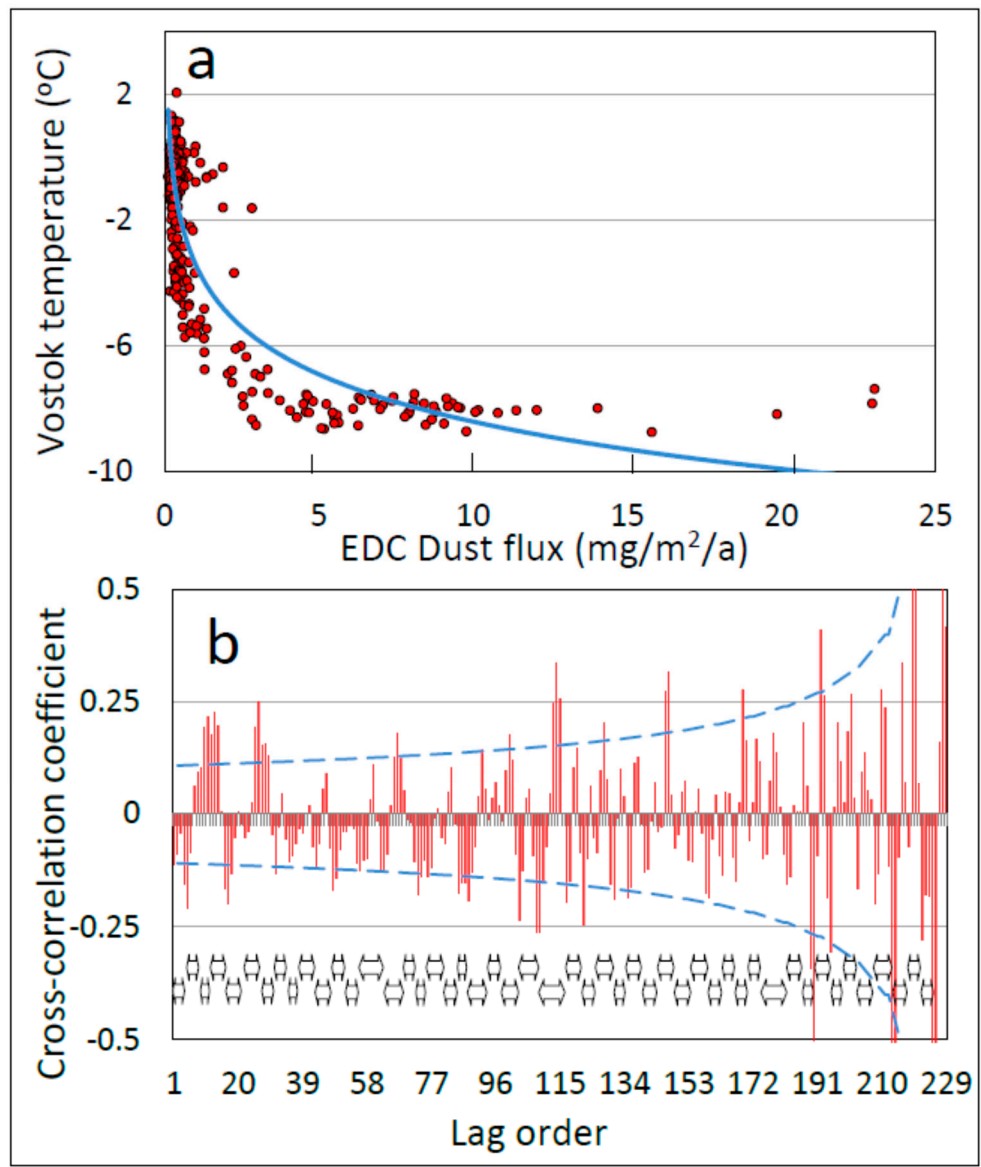

**Figure 10.** The relationship between dust flux at EPICA Dome C (EDC) and temperature proxies at nearby Vostok from the Last Glacial Maximum (LGM) to the late Holocene, for the time period 21,969–397 years before 1950 (Yb1950). (**a**) Scatterplot showing the logarithmic relationship. (**b**) Cross-correlogram of temperature and wind proxies. Horizontal open arrows designate visually identified centennial-scale periodicities in the cross-correlogram. The dashed blue lines in part b represent 95% confidence limits (positive and negative) for the value of the correlation coefficient associated with different lag orders (sample sizes).

### 3.3.3. The Last 75 Millennia

The negative correlation between temperature and windspeed established by proxies in Figures 10 and 11 is visible qualitatively in Figure 12 for a time period older by 40 millennia. Cross-correlation of wind- and temperature-proxy records reveals the familiar oscillation of cross-correlation coefficients against lag order at the periodicity of both cycles. Most peak values of cross-correlation coefficients are discernible at $p < 0.05$, demonstrating quantitatively the strong coupling between wind- and temperature-proxy cycles for this glacial time period. The same 1:1 coupling documented above between centennial-scale wind-proxy cycles at EDC and ACO/AAO temperature-proxy cycles at nearby Vostok occurs in the 28 temperature-proxy cycles from 72–63 Kyb1950 that correspond to the last major

glaciation (stadial) (Figure 12). We conclude that wind/temperature coupling characterizes both stadial and interstadial climates.

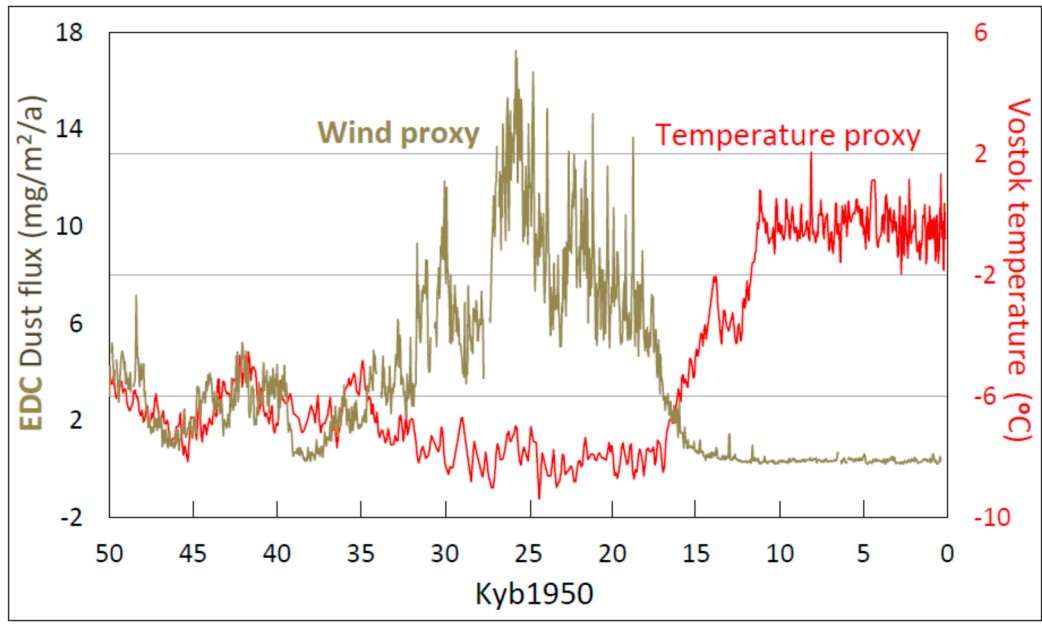

**Figure 11.** The relationship between wind- and temperature-proxies over the last 50 millennia showing positive correlation at low wind velocity and negative correlation at high wind velocity and during the exceptional calm during the Holocene.

The higher sampling resolution of the EDC dust-flux record combined with the 1:1 equivalence between major wind peaks and temperature peaks facilitates more accurate identification of potential AIM cycles from the wind-proxy record than from the temperature-proxy record, as illustrated in Figure 2. This finding supports the use of the wind-proxy record as a more accurate indicator of the integrated temperature record in Antarctica and globally. In this interpretation, the AIM cycle and the associated Bond cycle in the NH as reflected by D-O oscillations in Greenland ice-cores [35–38] are not confined to more recent paleoclimate record where these cycles were first identified, but instead recur on a regular cycle throughout at least the calmer periods of the last 226 millennia of climate history. In support of this hypothesis, we observe a strong and broad millennial-scale spectral density peak in both wind (Figure 13, peak at 1406 years) and temperature (Figure 13, peak at 1471 years) proxies (Figure 13). Consistently, spectral analysis of the wind- and temperature-proxy record over a nearby and largely-overlapping time period show similar spectral density profiles (Figure 13), as expected from the demonstrated tight coupling of wind-proxy and temperature-proxy cycles over this period (Figure 12). Matched spectral density peaks occur at the same frequencies ±0.5–2.8% on average (relative-absolute values). These findings demonstrate the need to reconsider the identification, analysis, and theory of distribution over space and time of the AIM (Bond, D-O, Heinrich) cycles based on the more informative Antarctic wind-proxy cycle, supplemented as possible and appropriate by the temperature-proxy cycle.

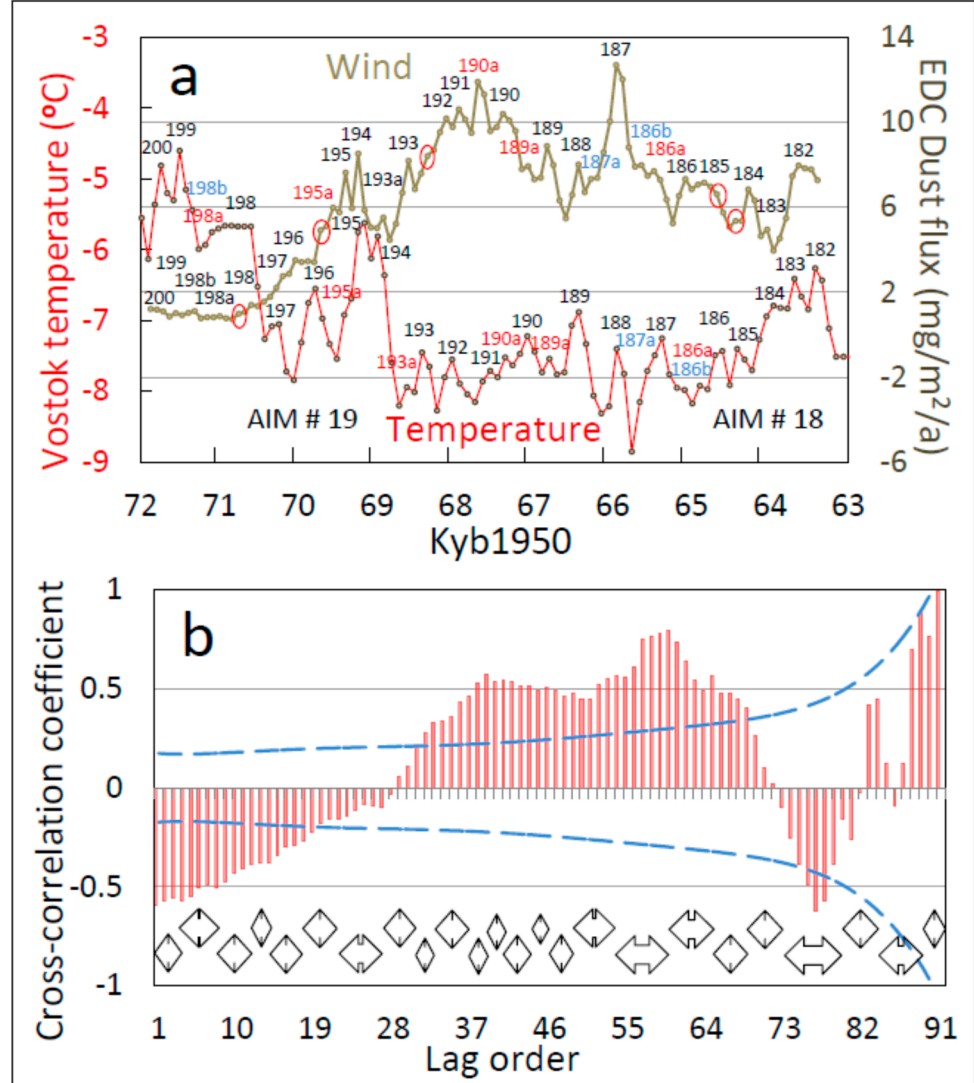

**Figure 12.** One-to-one coupling between proxies of wind at EPICA Dome C (EDC) and temperature at nearby Vostok during the last mid-glacial, 72–63 thousand years before 1950 (Kyb1950). (**a**) Time series showing 1:1 matching between wind peaks (brown curve) and temperature peaks (red curve) during the time of Antarctic Isotope Maxima (AIMs) 19 and 18. EDC temperature cycles are numbered following the original Vostok labeling nomenclature [12]. Additional possible AIM cycles (18a–18d, Figure 2) are not labeled in this graph. (**b**) Cross-correlogram between dust flux at EDC and temperature proxy at nearby Vostok for the time period 71.1–63.5 Kyb1950. The dashed blue lines in part (**b**) represent 95% confidence limits (positive and negative) for the value of the correlation coefficient associated with different lag orders (sample sizes).

We suggest above that AIM events are more frequent in climate history than previously believed (Figure 2) and find support for the hypothesis using spectral density analysis of Holocene climate data (Figure 9). Proxy time series data from much older and colder periods (Figure 13) likewise show strong spectral peaks in the millennial periodicity bandwidth, though in this case, the peak occurs at a period of 1406 years (wind) and 1471 years (temperature). These findings collectively support the hypothesis that putative cycles 18a–18d in Figure 2 are genuine and not artifactual. By analogy, and based on the wind record, these findings further support the hypothesis that AIM events are more frequent in climate history than previously known.

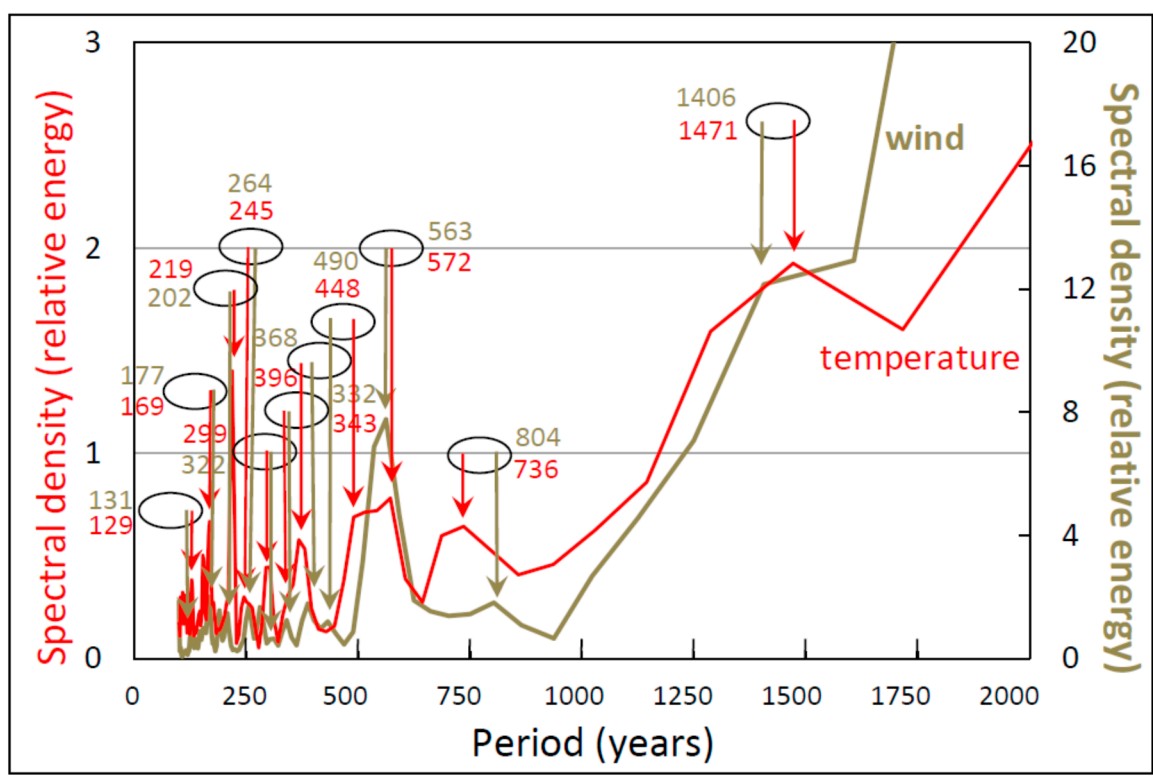

**Figure 13.** Spectral density periodograms showing similar spectral profiles of dust flux at EPICA Dome C (EDC, brown curve) and temperature proxy at Vostok (red curve) for the glacial period of ~75–65 thousand years before 1950 (Kyb1950). Ovals enclose shafts of arrows that identify the most closely matched peaks, with periods in years shown beside each oval. Fischer's Kappa statistic and corresponding probability that the distribution is discernibly different from white noise are 36.4, and $2.9 \times 10^{-7}$ (temperature record) and 84.10 and $p < 1.129 \times 10^{-65}$ (dust-flux record). The statistical discernibility of individual peaks was not determined, although by analogy with previous research [12] (Figure 3), the largest 6–9 peaks are probably significant at $p < 0.05$.

### 3.3.4. The Last 226 Millennia

The wind-proxy record over the last 226 millennia shows regular oscillations punctuated by quiet epochs corresponding to warmer interstadials (Figure 14, top panel). ACWOs are readily identified throughout this paleoclimate record during calmer periods, i.e., periods of lower wind velocities (Figure 14a–g). Four generalities emerge from broad comparison of wind- and temperature-proxy records.

First, the form of ACWOs is similar over the 226 millennia of the most recent paleoclimate record analyzed here. Wind-proxy cycles increase in amplitude over each millennium to culminate in a WT, following the ACWO waveform architecture observed during the Holocene (Figure 6) and as reported earlier for Vostok temperature-proxy records [12]. The duration of AIM cycles is apparently relatively unchanged at ~1000 y across the 226 Ky climate record analyzed here.

Second, wind-proxy cycles increase in peak amplitude over each millennial period (AIM cycle) by temporal summation and temporal facilitation, also reported earlier for temperature-proxy records [12]. Temporal summation and facilitation of wind-proxy cycles lead to and culminate in the WT, which paces the AIM cycle in the SH and corresponding Bond/Heinrich cycles in the NH.

Third, as noted above, the amplitude of wind-proxy fluctuations during the paleoclimate record is by far the smallest during the Holocene, demonstrating that in terms of wind velocity, the contemporary era is among the calmest in at least the last few hundred millennia of recent climate history.

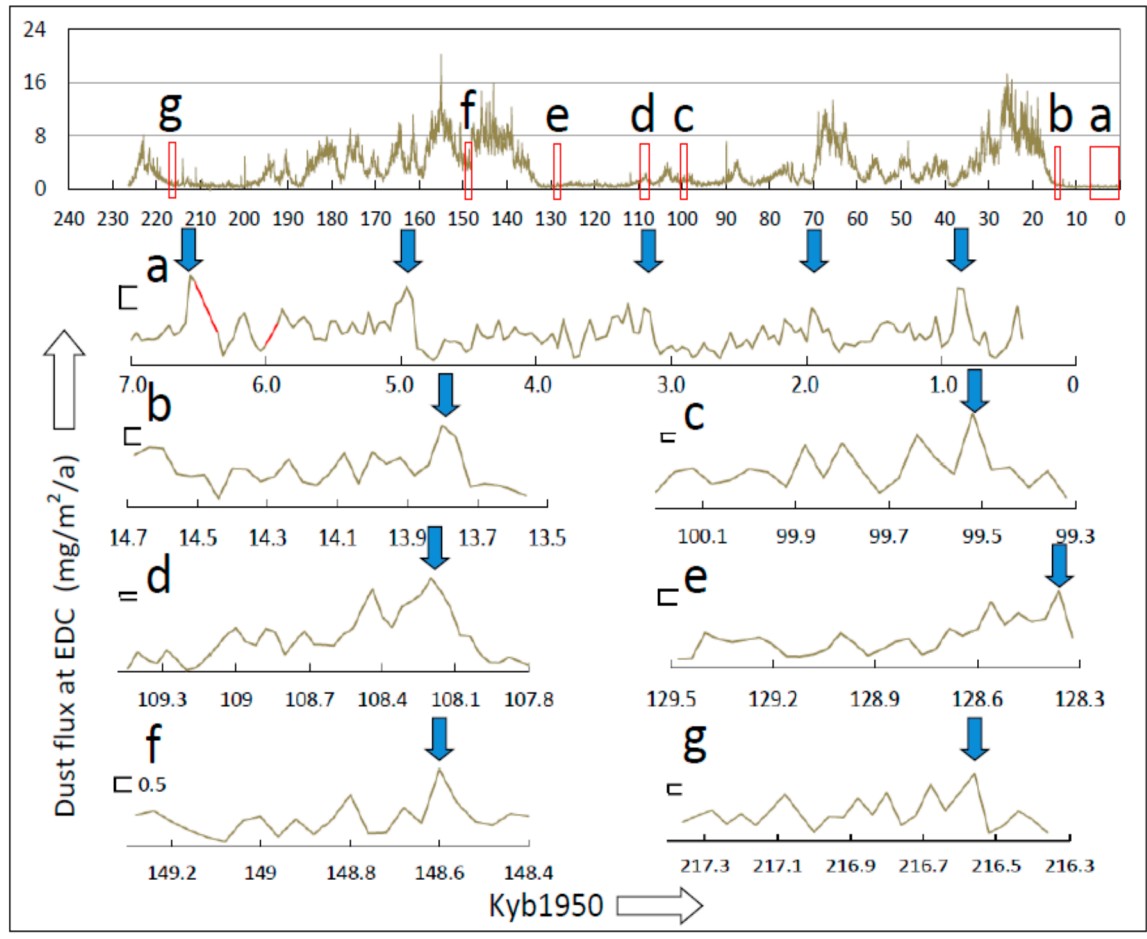

**Figure 14.** Time series of dust flux at EPICA Dome C (EDC) over the last 226 thousand years before 1950 (Kyb1950) showing the ubiquity and stereotypical form of the Antarctic Centennial Wind Oscillation (ACWO) on a millennial scale. Parts (**a**–**g**) represent time periods that correspond to the red rectangles labeled in the top panel. The calibration marks in (**b**–**g**) correspond to 0.1 mg/m$^2$/a of dust flux except in part f, where the scale is 50 times greater. Vertical blue arrows point to Wind Termini (WTs). The red segments of the wind-proxy curve at 6000-5900 Kyb1950 (panel (**a**) only) depict a period of missing wind-proxy data that were filled by interpolation for visual but not analytic purposes.

Fourth, the windiest period evaluated here was 149 Kybp (Figure 14, part f). Wind cycles of the ACWO appear clearly, but the amplitude of wind cycles is greater by a factor of more than an order of magnitude (note the different calibration scale in part f of Figure 14) than those in calmer periods. Most of the work reported here is confined to calmer periods of paleoclimate history, and this single exception 149,000 years ago does not enable generalities. This observation does suggest that the same ACWO cycle identified here primarily in calm periods of paleoclimate history also characterizes the windier periods, where the cycle is simply amplified manyfold.

## 4. Discussion and Conclusions

The primary result of this study is the documentation of a new Antarctic wind cycle, the Antarctic Centennial Wind Oscillation (ACWO), that can explain previously disparate climate phenomena and the consequent collapse of the largest 20 human civilizations of the last four millennia. We present evidence that these climate phenomena and civilizational milestones are driven by a common climate mechanism, cyclic modulation of wind-driven upwelling of heat and carbon dioxide from the depths of the SO forced by variation in surface wind stress associated with the ACWO.

Following this model, current global warming is interpreted as the peak of the contemporary ACO/AAO temperature cycle, which is coupled with and presumptively driven by the ACWO. In this interpretation, contemporary global warming is caused by the normal operation of the ACWO wind cycle and the ACO/AAO temperature cycle it forces, i.e., contemporary global warming is caused by a natural climate cycle.

Based on the historic pattern of the ACWO, a similar physical mechanism can explain several additional major climate milestones, including the MWP and LIA. The timescales of these climate milestones are longer than centennial-scale ACO/AAO cycles (three and six centuries, respectively), but these longer recurrent warm and cold periods are also coupled with increasing and decreasing winds of the ACWO cycle, respectively, and can therefore be explained by the same cyclic modulation of upwelling of heat and gas from the deeper SO. Similarly, AIMs, D-O events, and the Bond cycle result from the same forces. Cyclic modulation of upwelling in the SO by the ACWO therefore appears to be a general and universal mechanism underlying a broad array of climate phenomena.

If climate change follows the historic patterns disclosed in this study, the Earth has reached a centennial temperature maximum and global cooling is imminent. The ACWO architecture forecasts five decades of decreasing temperature followed by two or three additional ACWO cycles, steady net global warming corresponding to the next RWP, and in four centuries a recurrent little ice age, or RCP. The projected RCP is homologous with the LIA and previous RCPs that contributed to the decline of every major human civilization for the past four millennia. In this section we summarize some implications of these findings for climate science and policy and conclude with a more general discussion of necessary and sufficient conditions for causality as applied to the ongoing debate over the cause(s) of contemporary global warming.

## 4.1. Climate Mechanisms of the ACO/AAO

Our findings are consistent with the relaxation oscillation (RO) hypothesis of ACO/AAO generation developed previously [12,13], in which warming (cooling) triggers cooling (warming) (Figure 15a), based on the finding here that every ACO/AAO temperature cycle peak is accompanied by a centennial-scale ACWO wind cycle peak. In this interpretation of the RO process [13], the ACO/AAO temperature cycle is forced thermodynamically by the equator-to-pole temperature gradient. The polar temperature gradient in turn drives internal relaxation oscillation of Antarctic temperature resulting in alternating polar warming and cooling. This regional temperature oscillation is generated in the SO east of Antarctica, radiated across the Antarctic continent and projected northward to the remainder of the globe [12,13]. Well-established internal feedbacks are identified in support of each phase of the ACO/AAO cycle and initiation of the opposite phase [13] (Figure 15b). Some of these inputs and feedbacks are listed in the schematic RO model of Figure 15c.

To summarize the RO hypothesis of ACO/AAO generation (Figure 15c) [12,13], each warming cycle begins with increasing WWs associated with the ACO/AAO. Increased WWs enhance upwelling of Antarctic Intermediate Water (AIW) in the SO, accelerating the release of heat and $CO_2$ to ocean surface waters and the atmosphere and exporting colder surface water northward to lower latitudes by Ekman transport. As the warming phase of the ACO/AAO develops in Antarctica, the equator-to-pole heat gradient declines as a consequence, which reduces the intensity of Antarctic WWs. Slowing of WWs reduces wind stress at the surface of the SO, decelerates upwelling, lessens the northward export of cold water by Ekman transport, regenerates the Antarctic cryosphere, and induces the cooling phase of the ACO/AAO cycle. This cooling in turn increases the equator-to-pole temperature gradient, which re-energizes the WW cycles that drive the ACO/AAO temperature cycle to induce the next warming cycle. Well-established oceanic and atmospheric feedbacks are identified in support of each phase of the ACO/AAO temperature oscillation (Figure 15c). Conceptually related or similar climate mechanisms have been proposed or supported indirectly in several studies [12,13,39–44].

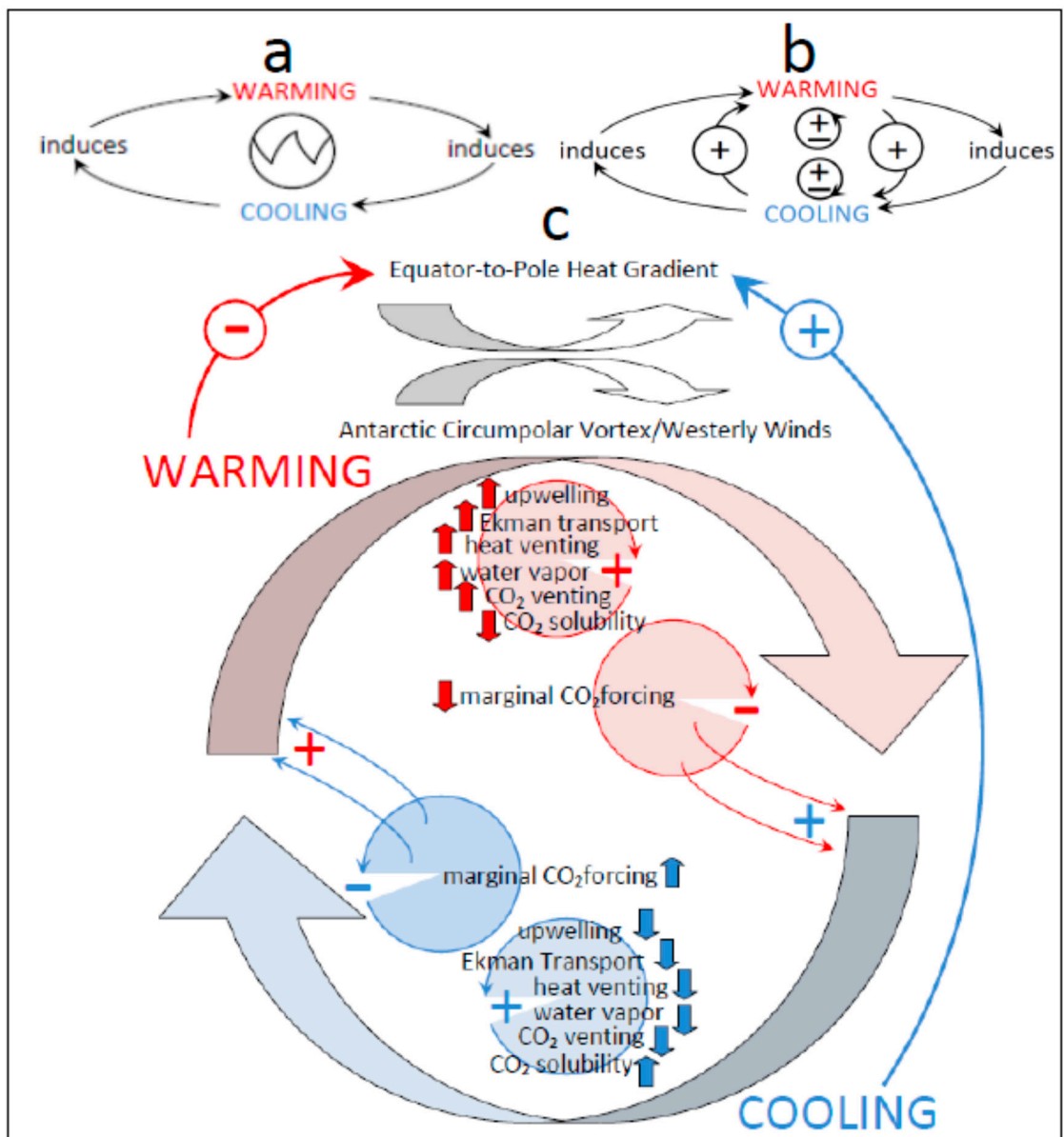

**Figure 15.** The relaxation-oscillation (RO) hypothesis of the Antarctic Centennial Oscillation (ACO) and its modern counterpart, the Antarctic Oscillation (AAO). (**a**) Schematic diagram of the ACO/AAO relaxation oscillator in its simplest form. (**b**) Schematic inclusion of the positive and negative feedbacks that trigger and reinforce each phase of the ACO/AAO temperature cycle. (**c**) Schematic summary of the RO model showing some but not all of the most significant internal and external feedbacks supporting the ACO/AAO temperature cycle. Plus and minus signs signify positive and negative feedback, respectively. This schematic diagram is a formalization of the RO model documented previously [13] and explains the ACO/AAO temperature cycle, and therefore global climate on centennial scales, in terms of relaxation oscillation of Westerly Winds (WWs) in the Southern Hemisphere (SH).

This and previous studies [12,13] therefore support a "systems" interpretation of the Antarctic climate, in which several air-sea-ice variables interact to produce the ACO/AAO temperature cycle. Internal feedbacks are understood to distribute causality throughout a network of interacting variables, in which case causality is mutual and reciprocal. By this reciprocity, wind causes temperature change, but temperature change also forces winds. In such cases of mutual causality within complex systems, some variables may nonetheless be more instrumental (have higher "gains") than others in forcing system output. We interpret the new wind cycle identified here, the ACWO, as the primary causal



variable underlying identified regional and global temperature cycles in part because the putative underlying mechanisms as summarized above, particularly wind-driven venting of heat and $CO_2$ from the SO, are so well documented and widely accepted.

*4.2. Antarctic Wind Oscillations Throughout Climate History*

Our study suggests that ACWO sequences terminated by WTs are ubiquitous during the last 226 millennia (Figure 14), as far back in time as centennial cycles can be resolved in the Vostok temperature-proxy record (see Section 2) [12]. AIM events and embedded ACWO cycles have recurred on a millennial cycle for at least the last 226 millennia, indicating that AIMs are more frequent in climate history than previously known. Future use of the more highly-resolved dust-flux record as a proxy for temperature [13] has the potential to extend this range from the 226 millennia studied here to the oldest time included in the EDC wind-proxy record [16], ~800 Kyb1950. These centennial-scale climate events are modulated on a millennial schedule and superimposed on the multi-millennial oscillation of the Great Ice Ages (GIAs), also termed Marine Isotope Stages (MISs). The ACWO wind cycle and the millennial-scale warm/cold ACO/AAO temperature cycles they force are the simplest repetitive unit and unitary building block of climate change on timescales most relevant to human and civilizational lifecycles, centuries and millennia. This paleoclimate cycle in Antarctica can be explained, both in timing and in mechanism, by the dynamics of the RO model formalized here (Figure 15) and the AIM cycle that is coupled with it.

The forces that govern the internal dynamics of these cycles during each AIM cycle have not been fully elucidated. Temporal summation of ACO/AAOs arises from algebraic summation of wind or temperature signals with repetition [12], but why do successive wind-proxy peaks within each AIM cycle grow larger with repetition by "temporal facilitation" to culminate in the WT? The same temporal pattern was reported in temperature records prior to the discovery here of the ACWO [12]. We speculate that each ACO/AAO cycle, and the consequent gradual warming of the Antarctic across each AIM cycle, conditions the Antarctic environment for the next and larger ACWO and the corresponding ACO/AAO cycle, culminating in the largest ACWO cycle of the millennial sequence, the WT. The WT is followed by a period of relative refractoriness, during which the Antarctic cryosphere is regenerated, followed by subsequent ACWO and ACO/AAO cycles that, in the newly refrozen Antarctic environment, are the smallest of the next AIM cycle.

Recent advances in understanding the mechanisms of the ACO/AAO [12,13] suggest a possible internal mechanism for temporal facilitation of cycle amplitude, namely, progressive reduction of the Antarctic cryosphere over each millennial AIM cycle. We hypothesize that such a long-term SH thaw distributed across successive ACWO wind cycles of each AIM cycle amplifies successive warming cycles to cause the observed increase in the frequency and amplitude of sequential ACO/AAO cycles. In this hypothesis, the regeneration of the cryosphere during each Recurrent Cold Period (RCP) slows subsequent warming and therefore limits the amplitude of early ACWO wind cycles in each AIM cycle. A shrinking cryosphere during successive ACWO wind cycles and corresponding warming cycles of the ACO/AAO implies that subsequent cycles warm faster, reach greater frequencies and amplitudes as observed, and culminate in the WT. The wind lull at the end of the WT induces cooling as described above, restores cryosphere volume, and therefore induces the same ACWO sequence in the next AIM temperature cycle as observed.

This hypothesis requires explanation of the increase in wind amplitude and velocity during successive ACWOs, as well as explanation of the increase in frequency of ACWO cycles during each AIM cycle. Both may be understood in term of bidirectional (reciprocal) feedbacks between the Antarctic cryosphere and wind and temperature cycles, which to our knowledge have not been elaborated. This hypothesis to explain the origin of temporal facilitation of ACWO wind cycles can be tested initially by evaluating the volume of the cryosphere in paleoclimate records over climate history to search for correlations with wind velocity and by elaborating and modeling possible mechanisms to explain how a reduced cryosphere might increase ACWO amplitude and frequency. Alternatively,

or additionally, the increase in ACWO amplitude over each millennial AIM cycle (temporal facilitation) may reflect the properties of external forcing, such as fluctuation in Total Solar Irradiance (TSI) and/or ultraviolet (UV) radiation caused by the millennial-scale Eddy solar cycle of sunspot frequency as discussed further below.

### 4.3. The Origin and Climate Mechanisms of the MWP/LIA

Both the MWP and the LIA have been described as "anomalous" temperature excursions, and the MWP is sometimes termed the "Medieval Climate Anomaly," e.g., [45]. Our study suggests that the warm/cold cycle that creates these temperature excursions is not "anomalous" in the sense of "exceptional or unusual," but is instead ubiquitous over at minimum the last 226 millennia of the most recent paleoclimate history (Figure 14). We therefore recommend replacement of the name "Medieval Climate Anomaly" with the most commonly-used alternative, the "Medieval Warm Period." Earlier and future manifestations of this natural global warming cycle are here termed "Recurrent Warm Periods" (RWPs), and identified by the corresponding AIM cycle number, i.e., RWP2 (Figure 6).

Substantial research on the MWP/LIA has been concerned with the geographic extent, e.g., [45], and variability [46,47] of these natural temperature excursions. Several investigators have concluded that "Increasing paleoclimatic evidence suggests that the Little Ice Age (LIA) was a global climate change event," [46] (p. 1223), although regional responses show significant variation in temperature anomaly across sites [48–50]. Several possible mechanisms to explain the origin of the MWP have been proposed, including solar variability [48,49], geomagnetic fluctuations on Earth [50,51], volcanic eruptions [48], responses to variation in radiative forcing from unspecified sources [45,52], changes in land-use patterns in Europe [53], and unspecified natural influences [54,55]. Previous work has shown further that these global temperature events are associated with shifting winds in the SH. Based on particle size distribution in the dust-flux record, it has been inferred that the WWs in the SH moved to a generally more southerly position from 1050–1400 CE, the approximate time of the MWP. At approximately 1430 CE, the approximate time of onset of the LIA, the WW belt shifted toward the equator [56]. Over the past half-century, the WW have shifted southward again and intensified [56], paralleling the contemporary AAO dynamic, which is now near its apogee.

Suggested mechanisms underlying the LIA include solar variability [55,57–61], volcanic eruptions [62–64], fluctuations in the Earth's magnetic field [50,65], periodic reversals of the solar magnetic field [66], and planetary orbital forces [67]. Here we describe a new climate mechanism for the MWP/LIA, driven or at least accompanied and hindcast precisely by the Antarctic winds identified in this study, the ACWO. By this mechanism the SO undergoes cyclic venting of heat and gases on millennial and centennial timescales, driven by the ACWO wind patterns documented in this study. The MWP is caused by the net warming that is forced across AIM cycles by steadily-building Antarctic winds. In this interpretation, the ACO/AAO grows over several AIM cycles to produce net warming at the site of ACO/AAO origin in the SO. This net warming signal teleconnects northward to manifest globally as the MWP. The same mechanism has operated for at least 226 millennia to cause Recurrent Warm Periods (RWPs), the evolutionary precursors of the MWP. As in the case of the contemporary global warming signal, warming during the MWP is interpreted as the result of net transfer of heat and gas from a venting SO forced by the intensifying ACO/AAO temperature cycle. The multicentennial warming recognized as the MWP therefore results from temporal summation and facilitation of successive ACO/AAO temperature cycles as described earlier for the formation of AIMs in the SH [12] (Figure 6).

The LIA that followed the MWP coincided with a deep lull in Antarctic winds. This lull represents the trough of the Wind Terminus (WT), the final ACWO wind cycle of the most recent AIM cycle. During this wind lull, the wind-driven upwelling that causes warming reaches its perigee, a millennial minimum that curtailed the venting of heat and $CO_2$ from deeper ocean waters to the surface and then to air and land while reducing the export of cold surface water to lower latitudes. These combined forces result in colder Antarctic temperatures and regeneration of the Antarctic cryosphere. In this

interpretation, the recurrent Antarctic cold period (the LIA and earlier RCPs) is caused by reduction of WW velocity and consequent cooling that originates in the SO [13], spreads across Antarctica [13], and teleconnects to the NH to cause the LIA and its thousands of historical precursors, here termed Recurrent Cold Periods (RCPs).

This simple physical mechanism based on the ACWO wind cycles and their variation across each AIM cycle therefore accounts for the full cycle of warm/cold temperature extremes that correspond to the MWP/LIA in the Antarctic and worldwide. Previous mechanism(s) proposed to explain the LIA, e.g., the series of volcanic eruptions at the beginning of the most recent LIA [62–64], may have reinforced the cooling induced by the corresponding WT and resultant LIA. Such volcanic events are probably not the prime mover of the LIA and its earlier homologs (RCPs) across all of paleoclimate history, however, except under the unlikely hypothesis that comparable synchronized volcanic eruptions occurred at the onset of each of the hundreds of RCPs that have occurred over the last 226 millennia.

Several previous studies have invoked internal influences such Earth's geomagnetic field fluctuation [50,51] or external influences such as the Sun [49–54] as a source of timing for multi-centennial climate fluctuations. We found previously that the repetition period of the ACO/AAO temperature cycle over the last 226 millennia declined on average by a factor of two, i.e., its repetition frequency doubled [12]. Substantially greater excursions in ACO/AAO period and amplitude occur during GIAs (MISs) and are rapidly reversible on centennial timescales [12]. Proportionate rapid and reversible centennial- and millennial-scale changes in solar output are unknown and implausible.

The solar deVries cycle (205-year period) is nonetheless reflected in climate records [58] and approximates the duration of ACWO and ACO/AAO cycles over the entirety of the Holocene (175 years) [12]. Moreover, harmonics of known solar cycles can in principle approximate the timing of paleoclimate temperature cycles [68,69]. For example, a climate model using the 87-year Gleissberg solar cycle and 205-year deVries periodicity generated a 1470-year Bond- and D-O-type cycles in response to freshwater inputs to the North Atlantic [68,69]. Recent studies have identified both centennial and millennial cycles of sunspot activity underlying the Maunder cycle [70]. The Antarctic wind regime documented in this study could therefore be forced by solar cycle harmonics generated by constructive and destructive interference, which are then low-pass filtered through the temperature inertia of the Antarctic icescape. Under this hypothesis, unmatched small wind peaks (e.g., red ovals in Figures 2 and 6) are sequences in a wind-driven train generated by relaxation oscillation, whose expression in heat balance is high-pass filtered by the temperature inertia of the Antarctic cryosphere. These considerations summarize one way the Sun could contribute to the timing of the ACWO and ACO/AAO.

The last decade has opened a new era in the study and prediction of solar activity on a millennial timescale [71]. These developments enable new insights into the relationship between solar activity and long-term climate events [72–75] such as the ACWO introduced here. Perhaps the strongest link between millennial-scale solar activity and climate is the Eddy Cycle of sunspot activity, which recurs on a ~1000-year cycle [72–74] and closely matches the timing of the ACWO cycle described here and corresponding fluctuations in terrestrial temperature. Low sunspot activity (e.g., the Maunder Minimum) correlates historically with low temperature on Earth, and hence the Eddy cycle, modulated perhaps by the longer Bray Cycle (2500-year period) [72–74], is a plausible source of the external energy fluctuation that drives the ACWO cycle described here. This hypothesis explains long-term recurrent warming (RWPs) and cooling (RCPs) in terms of increasing and decreasing TSI and/or UV caused by fluctuations in sunspot activity, mediated through the mechanism of the ACWO.

Our interpretations of global climate are based on the premise that major global climate milestones, such as D-O events and the MWP/LIA, originate in the SH and teleconnect to the north (South-to-North, or S-N teleconnection) [12,13]. Numerous previous investigators also concluded that D-O events in the NH originate from AIMs in the SH [24,35–38,76–81]. Other investigators suggested that interhemispheric causality is bidirectional under the bipolar see-saw model, e.g., [82,83]. Two studies, however, suggested that causality operates in the opposite direction, i.e., that D-O events and,

by analogy, other climate milestones, are forced southward from the NH [84,85]. Both of these studies, however, present empirical paleoclimate time series showing that AIMs in the SH began centuries to millennia before D-O events in Greenland ice cores ([84], their Figures 1 and 2b; [85], their Figure 1). Since cause precedes effect, data showing that AIMs in the SH are initiated before D-O events in the NH imply that the causal temperature increases originate in the south and propagate northward, consistent with the majority of studies demonstrating the primacy of the Antarctic in driving global climate events [12,13,24,35–38,76–81] and consistent with the present interpretations.

### 4.4. The Origin and Climate Mechanisms of the AIM, Bond and Heinrich Cycles

The data-driven climate model developed in this study based on the ACWO explains not only recurrent warm and cold cycles associated with glacial retreat and advance over the Holocene (MWP/RWPs and LIA/RCPs, respectively) but also the timing and the mechanisms of the AIM cycle. Variations in ACWO amplitude and frequency closely pace and indeed define the AIM cycle. Each AIM event comprises summated ACO/AAO cycles that are coupled with the ACWO in which each ACO/AAO cycle temporally summates or "piggybacks" on the previous cycle to reach a higher temperature that is followed after a variable northward teleconnection delay by D-Os in Greenland ice cores [12]. This study also identifies temporal facilitation of the ACWO wind cycle and accompanying ACO/AAO temperature cycles, in which individual warm/cold temperature cycles and corresponding ACWO wind cycles grow larger in amplitude over time and repetition to culminate in the maximum amplitude of the final ACWO cycle, the WT.

Most previous investigators have concluded that AIM cycles in the SH are the precursor of D-O events in the NH [12,13,24,35–38,76–81]. AIMs that originate in Antarctica are transmitted (teleconnected) to the NH to induce D-O events [12]. By analogy, the Bond cycle is the NH representation of the AIM cycle in the SH. Similarly, Heinrich ice-rafting events recurred on at least eight occasions during the Holocene centered on approximately 1400, 2800, 4200, 5900, 8100, 9400, 10,300, and 11,100 Ybp [86,87]. The dates of these NH events correspond in repetition frequency to the AIM cycle, and correspond in time to the early warming periods associated with the onset of RWPs in the SH (Figures 6 and 14). Heinrich events are caused proximately by the break-up of Arctic ice sheets that accompany warming [87]. Our findings suggest that the ultimate terrestrial cause of Heinrich events is recurrent thaws in the SH associated with the onset of warming that culminates in the MWP/RWPs. This hypothesis assumes rapid S-N telecommunication during the warmer Holocene, which has been demonstrated by others for interstadial climates [76,88].

We therefore suggest that the ultimate terrestrial cause of Bond and Heinrich events and cycles in the NH, as well as coupled D-O oscillations recorded in Greenland ice cores, is the ACWO wind oscillator in the SH that generates changes in temperature teleconnected across Antarctica [13] and globally. Regional teleconnection mechanisms in the SH include identified oceanic and atmospheric processes [13], which may be replicated on a larger scale to effect transmission of AIMs northward to induce D-Os [12].

In summary, our findings support a unified theory of global climate cycling on a centennial-to-millennial scale, driven by the ACWO and its variations within each AIM cycle. In this interpretation, the ACWO wind cycle is responsible for major climate milestones in both hemispheres, including the ACO/AAO temperature cycle in Antarctica, the AIM cycle that is marked by millennial summation/facilitation of ACO/AAO cycles [12,13], and the Bond cycle and related effects in the NH including D-O oscillations and Heinrich events. The ultimate external driver of these terrestrial cycles may be solar cycles such as the Eddy solar cycle of sunspot activity, in which cooler periods correspond to the millennial-scale Maunder Minimum or its earlier recurrent homologs.

### 4.5. Hindcasting Global Climate

A "gold-standard" for validating climate models is how well they hindcast past climate events identified independently [89–94]. We therefore compared ACWO hindcasts of the MWP and LIA

with empirically documented recurrent alternation of warm and cold periods associated respectively with the retreat and advance of glaciers during the Holocene. One empirical study reports six cold epochs centered at 8200, 6300, 4700, 2700, 1550, and 550 Ybp [57]. An independent empirical study documents six glacial expansions with similar timing, centered at 8.59–8.18, 7.36–6.45, 4.40–3.97, 3.54–2.77, 1.71–1.30 Kybp, and during the LIA [95]. Eleven of these twelve reported periods of glacial advance coincide with corresponding RCPs hindcasted by the ACWO wind cycle, including the most recent RCP, the LIA (Figure 6). The single outlier misses by a few decades. These comparisons suggest an empirical global climate hindcasting accuracy based on ACWO wind cycles of at least 95%.

Similarly, the ACWO wind cycle (Figure 6) hindcasts accurately the collapse of the largest and most prominent ancient human civilizations (Figure 7). Of the 20 largest human civilizations portrayed in Figure 7, all collapsed either at the beginning, during, or immediately following the LIA or its earlier RCP homologs, sometimes during a prolonged severe drought known to accompany such cold periods. The collapse of past civilizations is multifactorial [25–32], however, and climate change is identified here as one of several contributing causes. Our analysis here of the timing of all known civilizational starts and stops using data from 87 civilizations over the past 4000 years [25–32] (Figure 7) reveals that the rate of civilizational collapse during the LIA and homologous RCPs of the past is 2.56 civilizations/century, similar to the rate of 2.40/century during RWPs including the MWP. The rate of civilizational starts during the LIA and its recurrent homologs is 2.44/century, similar to the rate of 2.25/century during RWPs. This preliminary analysis therefore suggests that only the larger civilizations collapsed in association with recurrent cold eras (Figure 7) that accompany worldwide glacial advances, perhaps because of the greater burden of food security imposed by larger human populations.

*4.6. Forecasting Global Climate*

The ACWO wind cycle model developed here can be used to forecast global climate empirically following the extrapolation method pioneered by Liu and colleagues [96]. The present global climate forecasts are based on conventional methodology, consensually accepted climate (ice-core) data, and well-established oceanic and atmospheric processes and principles. Moreover, the forecasts offered here require only two major assumptions: first, the ACWO wind cycle identified in this study over the last 226 millennia will continue to operate for at least the next several centuries, the time equivalent of ~0.22% of the duration of the historical climate record analyzed here; and second, the possible counteracting effects of increasing $CO_2$ concentration on global temperature will be insufficient to arrest this natural temperature decline owing to the exponential reduction in marginal radiative forcing by $CO_2$ as its concentration in the atmosphere increases [7].

The climate forecast enabled by the ACWO wind cycle begins with the current state of the global climate. The ACO/AAO cycle is presently near or at its maximum positive (warming) phase and poised to cycle into its negative (cooling) phase within years to decades [12,13]. The ACWO-based forecast for immediate climate change is, therefore, the plateauing of Antarctic and global temperature followed by the decline anticipated to accompany the cooling phase of the current ACO/AAO cycle. This cooling is projected to propagate northward to the rest of the globe with little delay [76,88], as for D-O events [12], and teleconnect across Antarctica with little delay [13]. This forecasted cooling period can be expected to last for ~5 decades, the mean duration of the cooling phase of the last several ACO/AAO cycles. Imminent global cooling has been projected also by previous authors, e.g., [97,98].

After this cooling phase, Antarctic temperature is projected to increase again and oscillate near the mean recent period of the ACO/AAO cycle, ~100 years, for approximately three centuries. Following the ACWO pattern of the past, mean global temperature is projected to increase continuously during this period to manifest as the next RWP (i.e., the next MWP homolog). As in the past, the next RWP is projected to presage an extreme global temperature drop corresponding to the next LIA homolog, i.e., the next RCP, which is forecast to last for several centuries. The timing of the next RCP onset is estimated here simplistically using the duration of the most recent AIM cycle, 1080 years (Figure 6).

Since the LIA began ~660 years ago, the next RCP is projected to start in $1080 - 660 = 420$ years from the present, in the year 2440.

This projection is preliminary and incomplete, in part because it does not incorporate variance in ACO/AAO or AIM cycle amplitude or timing [12,13]. In particular, we cannot presently project the amplitude of the next cooling phase of the ACO/AAO temperature cycle with confidence. This limitation may be overcome by quantification of the RO model (Figure 15) and the ACWO wind-cycle (Figure 6) including computer and mathematical modeling that incorporates empirically measured variance in cycle amplitude and timing as a function of temperature and history. Such continued evolution of the empirical ACWO climate projection tool developed here may support more accurate and confident global climate forecasting based on consensually-accepted empirical climate data.

Despite the integrated character of the interactive climate system evaluated here, its elements can be identified and segregated for individual development, analysis, and simulation in the next-generation ACWO climate model. We recognize two main components of this model: the RO generator (Figure 15) and the ACWO (Figure 6). The RO generator explains the mechanisms underlying of individual ACWOs and accompanying ACO/AAO cycles on a centennial scale and is supported by the well-developed mathematical theory of relaxation oscillation [33,34,43]. The RO generator combined with the ACWO wind cycle and its variation within each AIM cycle together explain the behavior of observed trains of ACO/AAO cycles on a millennial scale. Coupled with heat and gas exchange provisions, such empirical models may be able to forecast a broad range of climate and civilizational milestones with unprecedented accuracy and time resolution.

### 4.7. Implications for Climate Science and Policy

Distinguishing conclusively between the two main competing hypotheses advanced to explain contemporary global warming, AGW and NGW, is a high scientific priority. The distinction will soon become self-evident, however. Whereas the AGW hypothesis forecasts continual warming, the NGW hypothesis as formulated here forecasts imminent global cooling as the current ACO/AAO cycle descends from the apogee of its present positive phase into the perigee of the coming negative phase. If and when global temperature begins a significant and sustained (decadal) drop despite increasing concentrations of $CO_2$ in the Earth's atmosphere, we can more confidently conclude that natural climate cycles are the predominant cause of global warming. The two contrasting hypotheses—AGW and NGW—are not necessarily mutually exclusive, however, and the relative attribution from natural v. anthropogenic sources to the contemporary global warming signal remains to be quantified.

In respect to climate policy, all extant climate policy instruments, whether local, regional, national or international, are based on the AGW hypothesis, which assumes human culpability for global climate change. If as we suggest here contemporary global warming results primarily from a natural climate cycle, rather than from human impacts on climate, then effective policy responses to this new empirically-projected climate scenario are limited to fault-free provisions that assume no human culpability for damages caused by global warming or cooling. Possible risks of elevated $CO_2$ to biodiversity remain to be established and integrated into policy [7].

Similarly, mitigation of natural climate cycles is both implausible and undesirable as a response strategy to natural climate change, except under the unlikely premise that humans can and should attempt to counter the massive forces that generate natural global warming. The only rational policy response to natural global climate change is adaptation to the imminent peak in global warming followed by adaptation to several decades of global cooling that are anticipated from the ACO/AAO temperature cycle. This climate scenario implies the need for revised incentive and compliance regimes to account for natural rather than, or at least in addition to, anthropogenic climate change. Limits on anthropogenic $CO_2$ emissions remain necessary and appropriate until it can be shown conclusively that ocean acidification from atmospheric $CO_2$ did not cause past mass extinctions [7] and poses no comparable future risk.

The last LIA persisted for nearly six centuries, causing mass starvation worldwide and the collapse of regional and global empires (Figure 7) [98]. The cause of these civilizational collapses was not human abuse of the environment nor collective societal choice [99], nor colder temperatures per se. Rather, civilizations apparently collapsed owing to the severe droughts that characterized the LIA and earlier RCPs, combined with wet, cool summers, which interfered with planting and harvesting. In particular shorter growing seasons prevented crop maturation and lowered or eliminated crop yields [100].

Given the nearly ten-fold expansion of the human population since the LIA, imminent global cooling and a projected recurrent LIA homolog in four centuries (the next RCP) could have far greater adverse effects on global agriculture, morbidity and mortality, mass population migrations, and political stability. Even the projected five decades of imminent global cooling forecast by the ACWO model will present new and unprecedented challenges to agricultural productivity. The corresponding new climate policy regime can be proactively and dynamically structured to help counter global food insecurity during imminent cooling and beyond to the deep global chill of the next RCP, which will be unprecedented since the last LIA.

### 4.8. Attributing Causality in the Global Warming Debate

Our study bears upon the cause(s) of the contemporary global warming signal and specifically on the AGW v. NGW debate summarized in the Introduction. In the AGW paradigm, contemporary global warming is attributed to anthropogenic greenhouse gases, especially $CO_2$, that have accumulated in the atmosphere as a consequence of human activities. In the NGW paradigm, contemporary global warming is attributed to the oscillation of one or more natural temperature cycles as identified in this and previous studies [7,12,13]. In weighing these alternative climate hypotheses, defining suitable criteria for causality is paramount.

The contrasting attributions of causality for global climate change can be adjudicated through the counterfactual framework of causation introduced by the Enlightenment philosopher David Hume [101,102]. In 1748 Hume wrote that:

> "We may define a cause to be an object followed by another, and where all the objects, similar to the first, are followed by objects similar to the second. Or, in other words, where, if the first object had not been, the second never had existed". [101] (Section VII)

As noted by others [102], Hume here erroneously conflated the second criterion of causality (necessity) with the first (sufficiency, his "temporal regularity"), but he nonetheless correctly identified what have become recognized as the two major criteria of causality, *sufficiency* and *necessity*.

The counterfactual formulation of causality has been elaborated for modern applications by the late David Lewis [103–105], among others (see [102] for an overview). To simplify, if event "a" (e.g., increasing atmospheric $CO_2$ concentration or a natural climate cycle) causes event "b" (e.g., global temperature increase), then event "a" must be both sufficient and necessary to the occurrence of event "b." Conversely, if event "a" is neither sufficient nor necessary to the occurrence of event "b," then event "a" is not the cause of event "b." In the current context, sufficiency is demonstrated if event "a"—either anthropogenic greenhouse gases or a natural climate cycle—can account for all or part of the contemporary global warming signal. Necessity is demonstrated if the same event "a" *must* occur in order for the same event "b" to take place in whole or part.

Within this counterfactual framework, consensually-accepted empirical data enable preliminary assessment of whether atmospheric $CO_2$ is causal—i.e., both sufficient and necessary—to contemporary global warming. Increasing atmospheric $CO_2$ concentration is not sufficient to cause significant temperature change because the marginal forcing power of $CO_2$ diminishes exponentially as its concentration in the atmosphere increases. By the start of the Industrial Age, marginal radiative forcing by $CO_2$ had already declined by half from the lowest atmospheric concentrations accompanying glacial maxima, and by now the marginal radiative forcing power of $CO_2$ has fallen to about one-third of

the maximum value reached during glacial cycling [7]. Owing to such diminishing returns, $CO_2$ that has accumulated in the atmosphere since the onset of the Industrial Age has had a minor but as yet unspecified impact on global temperature. As the concentration of $CO_2$ in the atmosphere increases further, the physics of the $CO_2$ molecule ensures that any further effect on global temperature will range from small to negligible [7].

Are changes in atmospheric $CO_2$ necessary to contemporary global warming? Recent research shows that for the past 425 million years, the concentration of $CO_2$ in the atmosphere is generally not discernibly correlated with global temperature [7]. The absence of correlation demonstrates the absence of causality. On the millennial scale of more recent glacial cycling over the last million years, atmospheric $CO_2$ concentration is correlated with temperature change, but available evidence suggests that $CO_2$ is not the cause. Instead, increased temperature occurs first and then atmospheric $CO_2$ increases, presumably because $CO_2$ is less soluble in warmer sea water and therefore is vented to the atmosphere from a warming ocean [8–11]. It follows that atmospheric $CO_2$ is not necessary to global temperature change. Since atmospheric $CO_2$ concentration is neither sufficient nor necessary to global temperature change, it is not causal.

The same criteria for causality can be applied to the natural temperature cycles that are the subject of the present study. If these natural climate cycles are causal to contemporary global warming as we propose, then they must likewise be both sufficient and necessary to the current global warming signal. Regarding sufficiency, and as developed in the Introduction to this paper, the ACO/AAO cycle exhibits orders of magnitude greater temperature-forcing power in the NH than is required to drive the smaller and slower +0.8 °C global warming signal registered since 1880. Moreover, the timing is appropriate to the sufficiency criterion: the current ACO/AAO cycle is at its maximum positive phase, i.e., the greatest wind stress of the last several centuries is occurring now across waters of the SO east of Antarctica where the ACO/AAO is generated [13]. Regarding necessity, the present study shows (Figure 14) that the ACWO wind pattern that drives the ACO/AAO temperature cycle has characterized paleoclimate over the past 226 Ky. While this observation does not alone prove necessity, it is reminiscent of Hume's "temporal regularity" criterion and consistent with the proposition that these natural climate cycles are necessary to the cyclic global warming that is now recurring.

Therefore, the counterfactual framework of causality supports the conclusion that the natural climate cycles identified here and in previous studies, the ACO/AAO [12,13] and the ACWO that drives it (this paper), are the primary cause of the contemporary global warming signal. Atmospheric $CO_2$ may be responsible for a small and declining fraction of the current global warming through its known greenhouse effects, but we conclude that the primary cause is the natural climate cycles explored here. The proportion of anthropogenic and natural contributions to contemporary global warming remains to be established.

**Author Contributions:** Conceptualization, W.J.D.; methodology, W.J.D., W.B.D; software, W.B.D.; validation, W.J.D., W.B.D.; formal analysis, W.J.D., W.B.D.; investigation, W.J.D.; resources, W.J.D.; data curation, W.J.D.; writing—original draft preparation, W.J.D.; writing—review and editing, W.J.D., W.B.D.; visualization, W.J.D.; supervision, W.J.D.; project administration, W.J.D.; funding acquisition, no funding. Both authors have read and agreed to the published version of the manuscript.

**Funding:** This research received no external funding.

**Acknowledgments:** This research was supported generally by the University of California at Santa Cruz and the Environmental Studies Institute, a non-profit (501)(3)(c) corporation based in Santa Cruz, CA, USA. We thank Jean Jouzel for kindly providing the EPICA Dome B database used in this study; Luke Kemp for generously providing the database on the timing of civilizational rise and fall; and Peter J. Taylor for inspiration, valuable discussion, and critiquing earlier versions of the manuscript. We thank several anonymous reviewers for comments that significantly improved the manuscript.

**Conflicts of Interest:** The authors declare that they have no conflicts of interest in respect to this paper.

## Abbreviations

| | |
|---|---|
| AAO | Antarctic Oscillation |
| ACO | Antarctic Centennial Oscillation |
| ACWO | Antarctic Centennial Wind Oscillation |
| AGW | Anthropogenic Global Warming |
| AIM | Antarctic Isotope Maximum |
| AIW | Antarctic Intermediate Water |
| °C | Degrees Celsius |
| CE | Current Era |
| $CO_2$ | Carbon dioxide |
| D-O | Dansgaard–Oeschger |
| $\delta^2H$ | Deuterium (temperature proxy) |
| EAP | East Antarctic Plateau |
| EDC | EPICA Dome C |
| GIA | Great Ice Age |
| Ky | Thousand years |
| Kyb1950 | Thousand years before 1950 |
| LD | Law Dome |
| LGM | Last Glacial Maximum |
| LGT | Last Glacial Termination |
| LIA | Little Ice Age |
| MWP | Medieval Warm Period |
| MIS(s) | Marine Isotope Stage(s) |
| N | North |
| NGW | Natural Global Warming |
| NH | Northern Hemisphere |
| ‰ | per mil |
| RCP | Recurrent Cold Period |
| RWP | Recurrent Warm Period |
| RO | Relaxation Oscillation |
| S | South |
| SH | Southern Hemisphere |
| SM | Supplementary Materials |
| SO | Southern Ocean |
| σ | Standard deviation |
| TSI | Total Solar Irradiance |
| UV | Ultraviolet radiation |
| v. | versus |
| WT | Wind Terminus |
| WW(s) | Westerly Wind(s) |
| Yb1950 | Years before 1950 |
| Ybp | Years before present |

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
