# Peer review of "Antarctic Winds: Pacemaker of Global Warming, Global Cooling, and the Collapse of Civilizations"

_climate, doi:10.3390/cli8110130_

Round 1
Reviewer 1 Report
Dear editor,
I found the paper by Davis and Davis interesting but many times not understandable.
I think that the authors need to improve their analysis methodology because the claimed results are not evident.
Below there are some suggestions, but I think that the analysis of the paper should be largely made again using a proper analysis. Also, the authors mix too many arguments. They need to better organize their work.
Abstract, line 29: why are you labeling the "Antarctic Oscillation" as "AAO" instead of AO?
Abstract, line 30: the abstract suggest that for you AAO is a "temperature". But the Antarctic Oscillation is defined with a pressure gradient and it is related to the movement of winds. Thus, it appears that your Antarctic Centennial Wind Oscillation(ACWO) and the Antarctic Centennial Oscillation.
Line 193: the authors state: "Lag order is a dimensionless number that designates the number of data points by which one record has 193 been shifted relative to the other in order to compute auto- or cross-correlations." This statement does not mean anything in particular if the data-points are sampled at different time intervals. Provide a properly defined time-lag analysis which must be accomplished by sampling the data at a constant time period. Then you can say that each step corresponds to a certain number of years.
Figure 1a. Why are you using Kyb2013? It is very odd. Use the common time definition. Are the dots equally spaced? How long is the period between consecutive dots? How did you connect the "cycles" using the dash-lines? To me the association appears totally arbitrary.
Figure 1b. Plot again the cross correlation function using a properly defined time-lag unit. As said above, using lag-order is meaningless if one does not know the time period associated with such order and, as said above, to use this methodology one needs to use records sampled at a constant time period.
Line 302: The authors states: "On this basis of this difference, we conclude that the cross-correlation between wind and temperature is 302 statistically discernible but weak despite the 1:1 matching of peaks." I am sorry, but I cannot discerne a 1:1 matching of peaks in your figure.
The same problems occur with all figures. The graphs are not readable because it is unclear what is the physical meaning of the "lag-order". Please use time-lags and plot all data figures using the common time definition instead of using kyb1950 and Kyb2013
Figure 9 and 13, plot the period using a log scale.
Figure 10a. I would say that the data do not follow the depicted log function at all. Try a function like y=a/x+b.
Figure 10b. Same problem. Lag-order is not comprehensible. Use time-lag.
Many other issues are present. But I could not follow the reasoning of the authors.
Reviewer 2 Report
I am satisfied with the revision and agree with publication in its present form.
Reviewer 3 Report
I am a priori inclined to believe this paper because it does seem that cold periods were bad for ancient civilizations. However, I have considerable difficulty with some of the figures (ie data) and assertions.
Fig 1: I do not see peak or cycle coherence and do not see cycles in either data set (temp or wind). Many of the dashed lines connect wind peaks with temp troughs and others the opposite. It is very hard to see that temperature leads wind, as they hypothesize. Fig 2: I fail to see a correspondence between wind and temp patterns at the large scale, even imagining lags. In both fig 1 and 2, the authors claim that firmly dated events enable reliable peak matching, but given data error and dating error in ice cores, how are we to assess the goodness of their peak matching which they admit is qualitative. They note up to a millenium latency from Law Dome to other sites. While antarctica is huge, this still seems hard to explain. Fig 4: more convincing peak match than earlier figs but still some peaks in wind match up with troughs in temp or are pretty flat. Peak 3 of temperature is the warmest of the 2000 years but lines up with the LIA. The entire LIA does not seem colder than the preceding MWP. This is not explained. Saying that 50% of cycles match up is not great and is certainly not 1:1 match. fig 6: It is stated that the MWP and LIA periods before the most recent ones are hypothesized based on the wind pattern. Ok fine but the periods do NOT line up with temp highs and lows at vostok. The AIM cycles are different lengths in the figure fig 7 civ collapse dates line up ok with LIA hypothesis but NOT with vostok temperatures. cases 2,3,6,7,9,10 line up with vostok temperature peaks not troughs. fig 9 with this many spectral peaks you need better period matches to declare good match, Need signif ones,visually the match is not good. fig 11 and 12 look ok fig 13 again dubious matching of periods. since this is all based on pattern matching, the LIA, MWP pattern is simply assumed from wind and defined rather qualitatively. The LIA and MWP type periods identified by the authors do not match my understanding of temperature reconstructions of the northern hemisphere and/or Europe or china. The would need to demonstrate that the wind pattern affected some aspect of N hemis temperature or rainfall for this all to make sense. Otherwise, how does antarctic wind affect civilizations in China? The other sets of matching peaks either are not as good as claimed or the authors need to clarify their methods.Author Response
please see attachment..

Round 2
Reviewer 3 Report
The authors categorically reject my criticisms. It is possible that I completely misunderstand what they did but I think not. Their explanation that the wind cycle teleconnects around the continent over a millenium seems like an excuse to justify failure of any drill site to match in time. Antarctica is about the same size as the United states, and while there are some differences in the PDO vs AMO effects on regional climate in different parts of the US, the country as a whole has coherent weather in the most part. Their claims about 100% peak matching do not make sense to me. I am fine if the editor wants someone else to review this but to say that no drill sites match up in time without being able to quantify that (what is lag around the continent) seems just hand-waving. And so on. sorry
Author Response
Reviewer Comments: The authors categorically reject my criticisms. It is possible that I completely misunderstand what they did but I think not. Their explanation that the wind cycle teleconnects around the continent over a millennium seems like an excuse to justify failure of any drill site to match in time. Antarctica is about the same size as the United states, and while there are some differences in the PDO vs AMO effects on regional climate in different parts of the US, the country as a whole has coherent weather in the most part. Their claims about 100% peak matching do not make sense to me. I am fine if the editor wants someone else to review this but to say that no drill sites match up in time without being able to quantify that (what is lag around the continent) seems just hand-waving. And so on. Sorry
Respectfully, the reviewer’s claim that we “categorically reject” his/her criticisms is not correct. In the original review, this reviewer made eleven specific criticisms. Of these eleven, we accepted two at face value, and made the recommended changes. We also explained in great detail why we were unable to accept the remaining nine criticisms, namely that they were incorrect. We nonetheless made changes in response to all eleven of the reviewer’s criticisms to avoid any possible similar confusion by future readers.
The majority of this reviewer’s previous criticisms were that “lag order” did not show units. That comment reveals this reviewer’s ignorance about our methods, which continues into this review. Anyone who understands auto- and cross-correlation knows that lag order has no units.
Future readers will have no problem with similar confusion because the auto- and cross-correlation methodology is now explicitly detailed in several places in the text, including the revised methods. The same methodology was used in all three of our previous Climate papers, passed muster with reviewers there, was accepted by the editors of Climate, and has never been criticized by anyone following publication.
By way of evidence of our responsiveness to this reviewer, I quote here from our original responses to Reviewer # 3, with our acceptance of the reviewer’s criticism highlighted in italics.
“Comment # 1: They claim that “ACWO wind peak precedes every temperature peak of the Antarctic Centennial Oscillation” using dust-flux in Ice core.[sic] However, I was not impressed by the evidence proposed in their graphical abstract because the claimed peak association appears very random. Sometime the peaks are simultaneous (e.g #13), by [sic] in many other cases they are not with shifts with even 1 or 2 centuries. Thus, they claim does not appear well established.
Response to Comment # 1: The reviewer is correct. The word "precede" was incorrect and has been changed. What is correct, more precisely, is that every temperature peak of the ACO/AAO temperature cycle is matched 1:1 with an ACWO wind peak. In the revised paper we show this for eleven drill sites in Antarctica, not just Vostok as in our original manuscript.
Wind peaks and temperature peaks occur at different times even though there is a 1:1 correspondence. We have altered this wording everywhere to account for this valid criticism. We also expanded our discussion of the methods to cover this point.”
The reviewer’s claim that we “categorically reject” his/her comments is therefore demonstrably false. We do not and did not reject this or any reviewer’s criticisms, as even a cursory review of our responses to reviewers indicates. We accept whatever criticisms we can, and make clarifications in the text in response to all other criticisms to avoid similar confusion by future readers.
This reviewer’s second series of criticisms mirror the first ones without acknowledging or responding to our detailed responses to the first set of criticisms. In particular, the reviewer again repeatedly calls for units to be added to “lag order” when in fact, as explained in both our responses to this reviewer and also in the paper itself, lag order does not have units.
The reviewer continues in this third review of our work that “Their explanation that the wind cycle teleconnects around the continent over a millennium seems like an excuse to justify failure of any drill site to match in time.” We do not offer “excuses” for any of our findings. We present replicable evidence based on consensually-accepted original data in support of every claim we make, and this reviewer has not criticized either the data we use nor understood the methods we use to evaluate them.
This is another careless misinterpretation by this reviewer. Nowhere have we stated that the wind cycle teleconnects around the continent over a millennium, since that is obviously false. What the paper states is quite the opposite, namely:
“Given the rapid movement of wind over long distances, we infer that the wind (dust-flux) signal at each drill site, including EDC used here, is similar or identical to the wind signal at every other drill site and at the source of the ACO/AAO in the SO [13]. [ll. 158-160 of Methods).
This reviewer’s comment therefore again betrays a fundamental misunderstanding of our work and a failure on the part of this reviewer to read our paper thoroughly enough to avoid such egregious errors.
It is not the wind cycle that teleconnects around Antarctica in a millennium as this reviewer misstates, it is the temperature cycle, which is repeatedly written throughout the text. We devote an entire previous scientific paper to demonstrating this key phenomenon, namely Climate, volume 7, issue 9, https://www.mdpi.com/2225-1154/7/9/112#. Here is the published abstract of that paper, with the key finding in dispute here highlighted by italics.
The Antarctic Centennial Oscillation (ACO) is a paleoclimate temperature cycle that originates in the Southern Hemisphere, is the presumptive evolutionary precursor of the contemporary Antarctic Oscillation (AAO), and teleconnects to the Northern Hemisphere to influence global temperature. In this study we investigate the internal climate dynamics of the ACO over the last 21 millennia using stable water isotopes frozen in ice cores from 11 Antarctic drill sites as temperature proxies. Spectral and time series analyses reveal that ACOs occurred at all 11 sites over all time periods evaluated, suggesting that the ACO encompasses all of Antarctica. From the Last Glacial Maximum through the Last Glacial Termination (LGT), ACO cycles propagated on a multicentennial time scale from the East Antarctic coastline clockwise around Antarctica in the streamline of the Antarctic Circumpolar Current (ACC). The velocity of teleconnection (VT) is correlated with the geophysical characteristics of drill sites, including distance from the ocean and temperature. During the LGT, the VT to coastal sites doubled while the VT to inland sites decreased fourfold, correlated with increasing solar insolation at 65°N. These results implicate two interdependent mechanisms of teleconnection, oceanic and atmospheric, and suggest possible physical mechanisms for each. During the warmer Holocene, ACOs arrived synchronously at all drill sites examined, suggesting that the VT increased with temperature. Backward extrapolation of ACO propagation direction and velocity places its estimated geographic origin in the Southern Ocean east of Antarctica, in the region of the strongest sustained surface wind stress over any body of ocean water on Earth. ACO period is correlated with all major cycle parameters except cycle symmetry, consistent with a forced, undamped oscillation in which the driving energy affects all major cycle metrics. Cycle period and symmetry are not discernibly different for the ACO and AAO over the same time periods, suggesting that they are the same climate cycle. We postulate that the ACO/AAO is generated by relaxation oscillation of Westerly Wind velocity forced by the equator-to-pole temperature gradient and propagated regionally by identified air-sea-ice interactions.
Figure 6 published in the above-referenced Climate paper, which is the primary basis for the claim highlighted above in the abstract in italics. The measured latencies are in years. Note in this figure that the reference latency at Law Dome of 0 years, and the maximum latency of 1,163 years at Taylor, i.e., millennial scale. These are quantified, replicable results that have been peer reviewed and published by Climate, which this reviewer now describes inaccurately as “hand waving.”
Remarkably, this reviewer then justifies his/her misinterpretation and misinformation by comparing Antarctica to the U.S., as follows:
“Antarctica is about the same size as the United states, and while there are some differences in the PDO vs AMO effects on regional climate in different parts of the US, the country as a whole has coherent weather in the most part.”
It is frankly bizarre to us that this reviewer ignores published work on Antarctica as documented above, and then suggests that the Antarctic climate should behave more like U.S. weather, and then use this false proposition as grounds for rejecting a scientific paper! It is beyond comprehension how any climate scientist could expect US weather to serve as a model for Antarctic climate. I am moved to wonder whether this reviewer understands such basics as the difference between weather and climate, and between polar and mid-latitude climate phenomena.
This reviewer continues his/her third review with this comment:
Their claims about 100% peak matching do not make sense to me.
The methodology that does not “make sense” to this reviewer is detailed in three previous papers published in Climate, and in the Methods section (lines 170-182), with reference to previous papers where the method is used, as follows:
"As in our previous studies [7,12,13], we compared different paleoclimate records, in this case temperature proxy and dust-flux records, using a standardized approach consisting of quasi-quantitative cross-record cycle peak matching followed by quantitative auto- and cross-correlation analysis (time domain) and spectral analysis (frequency domain). Labeling of Vostok temperature-proxy cycles began from the most recent cycle, which peaked 149 years before 1950 (Yb1950) and is labeled cycle # 1 (12). Peak matching of temperature cycles at Vostok with those at all other drill sites [12] was achieved by first identifying the nearest peaks in other temperature records and labeling them with the same cycle numbers as Vostok [12,13]. The use of "signpost" climate events whose time of occurrence is accepted consensually within the climate science community throughout the temperature-proxy record enables unambiguous cross-matching of temperature peaks in paleoclimate records from different drill sites [12,13]." (emphasis added)
What is it that does not “make sense” to this reviewer? He/she does not explain. We do not know how our methods be summarized any more clearly or simply. Why does this reviewer persist in such errors despite our identical responses to his/her previous reviews? Obviously, this reviewer is ignorant of the content of our previous papers (on which this one rests) and appears equally ignorant of the content of the paper under review.
We used exactly the same peak-matching method in all of our previous published Climate papers, and therefore the method clearly makes sense to the reviewers of those papers and to the editors of Climate, who accepted and published the papers. Why is this accepted methodology now being re-adjudicated by this reviewer? Why does this reviewer refuse to read or acknowledge our previous responses to the SAME criticisms? As stated in our previous responses, this practice is scientifically unjustifiable.
The reviewer concludes these most recent comments with the claim that “to say that no drill sites match up in time without being able to quantify that (what is lag around the continent) seems just hand-waving.
Once again, the reviewer appears to be criticizing results from our previous paper (op. cit.), which he/she obviously has not read in spite of our identical previous responses. We spent two years and a full scientific paper quantifying exactly that, and it is inappropriate to re-adjudicate that previous finding here.
This manuscript is a resubmission of an earlier submission. The following is a list of the peer review reports and author responses from that submission.
Round 1
Reviewer 1 Report
This paper provides interesting idea and some evidences about the key word of the global change, global warming. Although most evidences support that CO2 is the driver of current warming, it is not 100% for sure and always very good to have new ideas advance our understanding about the global system. As there are some evidences supporting this issue, I agree with publication after revision.
1 the first paragraph in the Introduction, it is not very needed to use the mismatches between multi-decadal variations of temperature and co2 to be against the idean that co2 drives warming. Most scientists agree that CO2 drives the trend of the warming and multi-decadal oscillations of temperatures are driven by oceanic and atmospheric modes such as amo and pdo.
2 introduction, you have many terms that seems proposed from your previous studies. Most people know about AAO and D-O but not others. Can you explain the differences between them and state your purpose in a very clear way? As you have published two papers in the journal about this issue, it is necessary to clear state the advances.
3 line100, the abbreviation for D-O have existed before. There are also other abbreviations such as RO have the similar problem, please check through the manuscript.
4 I feel the authors used too many abbreviations. You may just use abbreviations for some key terms.
5 I strongly suggest the author to largely revise the manuscript to make it more readable. You can focus on the key parts, such as what are the new ideas and evidences. Many scientists proposed the winds driving temperature changes over Antarctica. What are the new ideas you added? You may provided evidences that previous one did not provided. Then interpret that why you think you idea is more robust. What are the new things from you previous two studes. You have repeated many issues and there is no need to mention about some not important issues. For example, there is no need to mention about changes of the global warming and repeated warming or so. This works if your ideas are well accepted naturally. Most people agree with that the lia occur repeatedly also. The paper is too long with too many not important issues. I am very interested in your study and feel it is not easy get your key points.
Reviewer 2 Report
I don't understand why the authors are using temperature records from Vostok and dust records from Dome C, rather than the temperature records from Dome C. The resolution appears to be similar for both cores, and correlating the dust from Dome C to the winds from Vostok seems like a completely unneccessary extra step that introduces uncertainty into the record. Since all of this hinges on fairly short cycles relative to the resolution in these ice cores, you'd think the authors would want the best possible match between records, rather than correlating two separate cores with only the justification that the much lower resolution Vostok dust record is "remarkably similar" to the higher resolution Dome C dust record. Remarkably similar they may be at the 600 year resolution Vostok provides, but we can't be certain that Vostok looks exactly like Dome C in dust at higher resolution. Since Dome C has both high resolution dust and temperature, I don't understand at all why they didn't just use that. I think if the authors did that, and took out all the grandiosity about how this one wind pattern controls all of global climate and human civilization history, it might be more worth reading.
Reviewer 3 Report
Dear Editor,
I found the work by Davis & Davis potentially interesting, but I believe that the authors need to support their conclusions better.
They claim that “ACWO wind peak precedes every temperature peak of the Antarctic Centennial Oscillation” using dust-flux in Ice core.
However, I was not impressed by the evidence proposed in their graphical abstract because the claimed peak association appears very random. Sometime the peaks are simultaneous (e.g #13), by in many other cases they are not with shifts with even 1 or 2 centuries. Thus, they claim does not appear well established.
Figure 1A presents the same random peak association of the graphical abstract.
Figure 1b should add the units of the lag order.
Figure 2 shows again the same problem of figure 1. This kind of graphing is really not convincing because it is always possible to take two fluctuating sequences and then connect randomly the peaks.
Figure 3 has the same problem of figure 2, and again the units are missing in B.
Figure 4 and 9 are again ambiguous. Two power spectra are compared but again the presumed coherence is left to a visual impression that again I could not see well.
Figure 5 and 6 is again not readable because the units are missing.
The other claims from page 26 (section 4. Discussion and conclusion), lack the scientific support of the previous sections and are too qualitative.
Part of the ideas of the authors may be correct. However, the analusis methodologies that they have used to support them are very inadequate.
For example, if they claim a spectral coherence between two records, they should use spectral coherence methodologies. Relative phases can be determined with wavelet cross spectra, ecc. Without a sufficiently robust analysis, their work appears too vague. Even the simple visual correlation that the authors propose are ambiguous and arbitrary because of the apparently random peak association.
Thus, I suggest the authors to spend more time in improving their research.